# Diversity Measures for Niching Algorithms

**Jonathan Mwaura** [1,*]**, Andries P. Engelbrecht** [2] 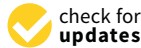 **and Filipe V. Nepomuceno** [3]

[1] Department of Computer Science, The University of Massachusetts Lowell, Lowell, MA 01854, USA

[2] Department of Industrial Engineering and Computer Science Division, University of Stellenbosch, Stellenbosch 7602, South Africa; engel@sun.ac.za

[3] Merlynn Intelligent Technologies, Pretoria 0140, South Africa; filinep@gmail.com

[*] Correspondence: jonathan_mwaura@uml.edu

**Abstract:** Multimodal problems are single objective optimisation problems with multiple local and global optima. The objective of multimodal optimisation is to locate all or most of the optima. Niching algorithms are the techniques utilised to locate these optima. A critical factor in determining the success of niching algorithms is how well the search space is covered by the candidate solutions. For niching algorithms, high diversity during the exploration phase will facilitate location and identification of many solutions while a low diversity means that the candidate solutions are clustered at optima. This paper provides a review of measures used to quantify diversity, and how they can be utilised to quantify the dispersion of both the candidate solutions and the solutions of niching algorithms (i.e., found optima). The investigated diversity measures are then used to evaluate the distribution of candidate solutions and solutions when the enhanced species-based particle swarm optimisation (ESPSO) algorithm is utilised to optimise a selected set of multimodal problems.

**Keywords:** diversity; niching; multimodal optimisation; particle swarm optimisation

## 1. Introduction

For population-based search algorithms such as genetic algorithms (GA), differential evolution (DE) and particle swarm optimisation (PSO), population diversity (note that, for PSO, the term swarm diversity is preferred as the particles (candidate solutions) are said to be part of a swarm. The rest of the paper uses PSO specific terminology.) provides an indication of the amount of exploration or exploitation done in the population/swarm. High diversity means that the solutions are dissimilar which implies that the algorithm is still exploring the search space. Conversely, low diversity means that the algorithm is converging to a solution [1].

Diversity measures are used to provide insight on the search state, i.e., exploring or exploiting [2], as well as to provide information on how an algorithm spreads its candidate solutions within the search space.

Multimodal optimisation involves locating all or most of the optima present in a multimodal problem. These optima could be both local or global. The objective of multimodal optimisation techniques is to find as many of these optima as the designer might find interesting and satisfying for a particular problem [3,4].

For a niching algorithm, a high swarm diversity may either imply a wide exploration of the search space or a diverse set of found optima (or niches), here referred to as solutions. As such, while investigating diversity for niching algorithms, it is important to make distinctions between:

- **Swarm diversity** which refers to the diversity with respect to the decision space, i.e., particle positions of the current swarm.
- **Niche diversity** which refers to the diversity with respect to the solution space, i.e., actual found solutions. Niching algorithms obtain multiple solutions by forming clusters of a few particles around the positions of optima. The cluster of particles is

referred to as a niche while the fittest particle in the niche, the so called neighbourhood best (nbest), represents an optimum. Niche diversity refers to the spread of these neighbourhood bests in the solution space.

- **Phenotypical diversity** which refers to the diversity with respect to the objective space, i.e., the diversity with respect to the objective function values of the current particles or the nbests (depending on the research interest in question).

The review and empirical study reported in this paper focus on the first two diversity criteria, in the context of niching algorithms.

The main goal of this paper is to review diversity measures and to propose a set of diversity measures for niching algorithms. As such, the paper does not include a review of PSO and other complimentary algorithms. In addition, whereas the experiments make use of the enhanced species-based PSO niching technique (ESPSO) [5], this paper does not revisit niching algorithms in general or the ESPSO in particular. The ESPSO niching algorithm is only used to illustrate the capability of the diversity measures to quantify particle and niche dispersion in the search space. The reader is directed to find a comprehensive review of PSO niching algorithms in [6,7].

The rest of this paper is organised as follows: A sufficient review of diversity measures is presented in Section 2. The diversity measures are used in Section 3 to quantify the dispersion of particles and niches when the ESPSO is utilised to optimise a selected set of multimodal problems. Section 4 provides a discussion of the results. Conclusions are drawn in Section 5.

## 2. Diversity Measures Utilised in Population Based Algorithms

In this section, diversity measures' developed for single solution (non-niching) population-based algorithms are analysed for their applicability as diversity measures for niching algorithms. Where the discussed measures' applicability is not sufficient for niching algorithms, modifications are proposed to make them suitable for niching algorithms.

Throughout the paper, the following terminology is used:

- Swarm, $S = \{\mathbf{x}_1, ..., \mathbf{x}_i, ...\mathbf{x}_n\}(2 < n < \infty)$, which refers to all particles in a PSO algorithm. In other complimentary techniques such as a GA, $S$ refers to the population.
- Candidate solution, $\mathbf{x}_i$, which refers to a particle in a swarm, $S$. A candidate solution is incrementally adapted to find an optimum.
- Candidate niche, $N_k = \{\mathbf{x}_i, ..., \mathbf{x}_m\}(m \leq n)$, which is formed by a cluster of candidate solutions. A candidate niche can still be refined to find an actual solution, or optimum.
- Neighbourhood best (nbest), $\hat{\mathbf{x}}_i \in N_k$, which is the particle with the best fitness in a candidate niche. This then represents an optimum.
- Solution(s): The final neighbourhood best or optimum (i.e., particle with the best fitness) in a particular identified niche.

The remainder of this section focuses on swarm diversity measures in Section 2.1 and niche diversity measures in Section 2.2.

### 2.1. Swarm Diversity

Diversity with respect to the decision space, i.e., swarm diversity, aims to quantify how an algorithm spreads candidate solutions (particles) in the search/decision space. This section reviews swarm diversity measures. In addition, the discussion focuses on how the reviewed measures can be used to quantify spread in niching algorithms.

#### 2.1.1. Sum of Distances

The sum of distances (SD) measure finds the dissimilarity between candidates solutions, by calculating the square root of the sum of their distances from one another [4]. Equation (1) shows how this is computed:

$$D_{\text{SD}} = \sqrt{\sum_{\mathbf{x}_i \in S} \sum_{\substack{\mathbf{x}_j \in S \\ j \neq i}} \left\| \mathbf{x}_i - \mathbf{x}_j \right\|_2} \tag{1}$$

where $S$ is the set of all candidate solutions at a given time, while $\mathbf{x}_i$ and $\mathbf{x}_j$ are different candidate solutions in $S$.

Over time, niching algorithms drive particles to cluster around optima (or niches). Thus, at convergence of a niching algorithm, particles will be clustered around a niche, with the best particle of a niche (i.e., the neighbourhood best) representing one of the found optima. Figure 1a illustrates the spread of particles during a search process, while Figure 1b illustrates the convergence of particles, in the form of niches, around positions of optima. Appendices A–C show progression of particles from initial (random) positions as well as their positions at different iterations during a typical run.

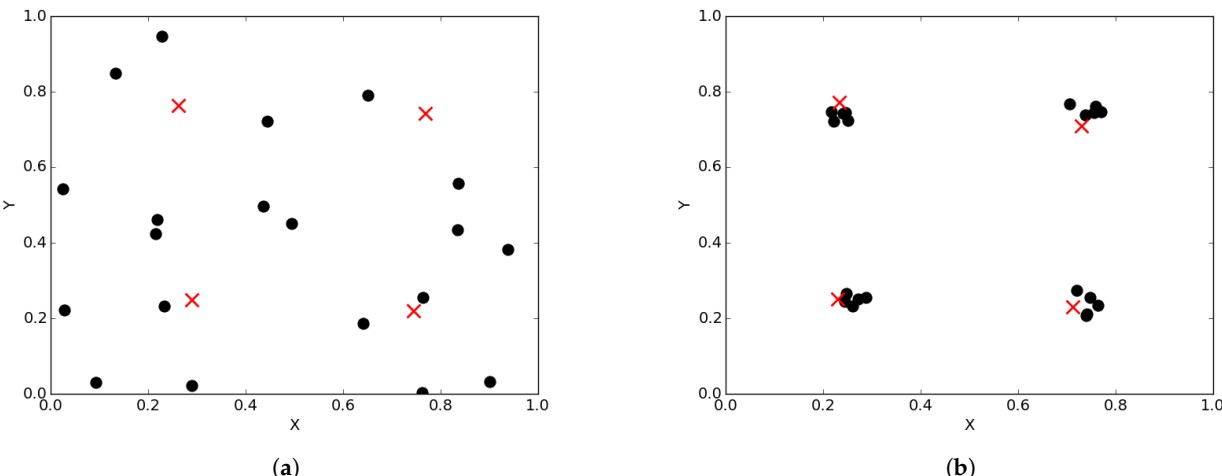

(**a**)　　　　　　　　　　　　　　　　　　　　　　　　　　(**b**)

**Figure 1.** Illustration of swarm diversity for niching algorithms. The crosses show the positions of optima. (**a**) particles during search process; (**b**) particles at convergence.

The expected behaviour in niching algorithms is that, when candidate solutions start converging near their neighbourhood bests, intra-niche (distances between particles in the same niche) distances start to approach zero. This means that the swarm diversity at convergence should approach zero. For the case of the SD measure, the expected behaviour is that even at convergence, calculated SD values will be high, in particular if the inter-niche (distances between the best particles of the niches) distances are large. Thus, calculated SD values gives an incorrect impression of diversity at convergence. Figure 2 depicts how swarm diversity compares to inter- and intra-niche distances over time.

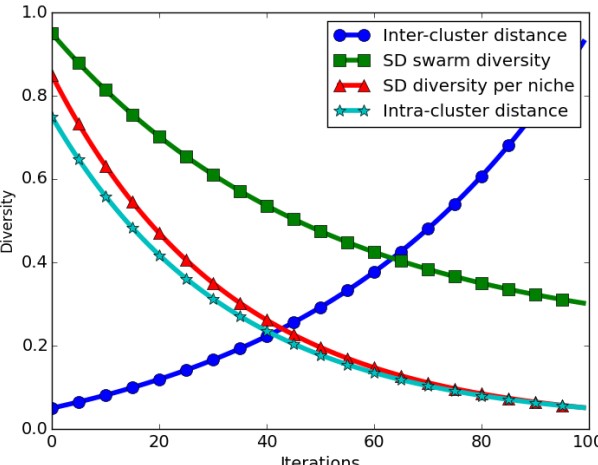

**Figure 2.** Illustration of the expected diversity values when quantified using the SD measure.

In order to adapt the SD measure to correctly quantify spread for a niching algorithm, the SD measure should be computed for each identified candidate niche. The values of the SD measure for each of the candidate niches are then summed and averaged. The modification of Equation (1) to calculate diversity for a niching algorithm is then given by

$$D_{mSD} = \frac{1}{\mu} \sum_{k=1}^{\mu} \left( \sqrt{ \sum_{\mathbf{x}_i \in N_k} \sum_{\substack{\mathbf{x}_j \in N_k \\ j \neq i}} \| \mathbf{x}_i - \mathbf{x}_j \|_2 } \right) \tag{2}$$

where $\mu$ is the number of candidate niches, $N_k$ is the set of particles in a candidate niche, $k$, while $\mathbf{x}_i$ and $\mathbf{x}_j$ are different candidate solutions in $N_k$.

Computing the swarm diversity as an average of the candidate niches' diversity eliminates the inter-niche distances problem reported earlier. The illustration shown in Figure 2 depicts that, as niches converge, the average SD over all niches approaches zero. As shown, an averaged candidate niches' diversity (SD diversity per niche) corresponds to the intra-niche distances, which is the expected behaviour for a niching algorithm.

### 2.1.2. Sum of Distances to Nearest Neighbour

As the name suggests, the sum of distances to the nearest neighbour (SDNN) [8] calculates diversity of candidate solutions by summing up the pairwise distances of each candidate solution to its closest neighbour [9]. The SDNN is calculated as

$$D_{SDNN} = \sum_{\mathbf{x}_i \in S} \min_{\substack{\mathbf{x}_j \\ j \neq i}} \| \mathbf{x}_i - \mathbf{x}_j \|_2 \tag{3}$$

While it is reasonable to assume that a pair of nearest neighbours (NN) are particles in the same niche, it is still possible that a candidate solutions' nearest neighbour is a particle in a niche that has already converged. This may lead to the calculated $D_{SDNN}$ values being high even if niches are converged. As a result, the diversity computed using the SDNN measure may not be as expected during convergence.

For the SDNN measure to be applicable for niching algorithms, the diversity should be calculated for each candidate niche, i.e., the NN of each particle is selected from that particle's niche. The obtained values should then be summed and averaged to give one SDNN score for the swarm. The adapted SDNN measure is computed as

$$D_{mSDNN} = \frac{1}{\mu} \sum_{k=1}^{\mu} \left( \sum_{\mathbf{x}_i \in N_k} \min_{\substack{\mathbf{x}_j \\ j \neq i}} \| \mathbf{x}_i - \mathbf{x}_j \|_2 \right) \tag{4}$$

The adapted measure ensures that the computed $D_{m\text{SDNN}}$ value is a measure of the sum of intra-niche nearest neighbours distances.

Niching algorithms define niche formation differently. For instance, the ESPSO algorithm [5] utilises the concept of different species in a population to form niches. Broadly, the algorithm starts by firstly identifying species seed which form the candidate niches($N$). For the ESPSO, each identified species seed is the best fit particle in a neighbourhood, i.e., the nbest. Species seed is determined by initially sorting all particles, $x_i$ in the main swarm, in decreasing order of fitness. In addition, a parameter $r_s$ is defined which is the Euclidean distance between the species seed and the boundary of the species (i.e., the candidate niche($N$)). Particles that fall within $r_s$ distance from the species seed are part of the species. If a particle is found that falls outside the radius of that seed, then it is also added to the set $N$ as a new species seed. As N continues to be filled, subsequent particles are checked against all seeds $N$ adding only those particles that do not fall within any radii of any of the seeds in $N$. The set of species seed is complete once all particles have been checked. This means that in ESPSO each particle is part of a distinct specie (candidate niche).

Unlike the ESPSO, in other PSO niching algorithms, particles could still belong to a particular niche even if their Euclidean distance from their neighbourhood best is somewhat large. As a result, while a high SDNN value means that particles in a niche are far from each other, a low SDNN value may not indicate niche convergence because the calculated distance is only with respect to NNs and not entire members of a candidate niche. Thus, the modified SDNN measure is affected by how a niching algorithm defines niche membership. A potential solution to this problem is to calculate the average SDNN for each niche and then to average over all niches. In essence, this will yield an average intra-niche NNs distance. However, for the purpose of this work, the $D_{m\text{SDNN}}$ measure was calculated as per Equation (4).

### 2.1.3. Average Distance around the Swarm Centre

The average distance around the swarm centre (ADSC) computes the swarm diversity by calculating the average distance of particles $\mathbf{x}_i$ to a central point $\bar{\mathbf{x}}$ in each dimension [10,11]. The central point, $\bar{\mathbf{x}}$, can be either the average over all particles or the best particle in the swarm. This diversity measure is calculated as

$$D_{\text{ADSC}} = \frac{1}{n} \sum_{\substack{i=1 \\ \mathbf{x}_i \in S}}^{n} \|\mathbf{x}_i - \bar{\mathbf{x}}\|_2 \tag{5}$$

where $n$ is the number of particles in the swarm $S$.

A small value of the ADSC indicates convergence around the centre of the particles, while a larger value means that the solutions are scattered around the centre.

If the ADSC was to be used to compute the swarm diversity for niching algorithms using a central position or a best particle position, an incorrect niche diversity trend will be illustrated. This is because of the large distances that are likely to exist between a central point and a candidate niche or between the global best position and the candidate niches. As such, the ADSC measure has to be adapted in order to be applied to niching algorithms.

The ADSC measure is adapted for niching algorithms by using the neighbourhood best in a candidate niche as the reference point for all particles in that candidate niche or using an average point over all particles in a candidate niche. For the reported work, the neighbourhood best was used. That is, in Equation (5), the term $\bar{\mathbf{x}}$ refers to the neighbourhood best position in the candidate niche to which particle $\mathbf{x}_i$ is a member. Using this adaptation, ADSC is a suitable diversity measure to quantify diversity for niching algorithms because it now computes the average distances of each particle to its neighbourhood best.

2.1.4. Average of the Mean Distance around All Candidate Solutions

The average of the mean distance around all candidate solutions (ADAA) is used to measure the average distance between each pair of particles [11]. This measure is calculated as

$$D_{\text{ADAA}} = \frac{1}{n} \sum_{i=1}^{n} \left( \frac{1}{n} \sum_{\substack{\mathbf{x}_i \\ \mathbf{x}_j \in S \\ j \neq i}}^{n} \| \mathbf{x}_i - \mathbf{x}_j \|_2 \right) \tag{6}$$

This measure shares similar characteristics with the SD measure. The only difference is that an average distance of a candidate solution from all other solutions is calculated and an overall average distance is computed. As such, when utilised for calculating diversity for niching solutions, ADAA will encounter similar drawbacks as the SD measure.

To adapt the ADAA measure for niching algorithms, an ADAA diversity value is computed for each candidate niche. The values are then summed for all candidate niches and an average calculated. This ensures that the result implies the average of the mean distance around all candidate solutions in each candidate niche.

2.1.5. Entropy

Entropy is defined as a measure of the uncertainty involved in choosing an element from a source according to some probability distribution [12]. In the context of diversity, a high entropy measure value means a high diversity index [13–17]. In the work presented here, the entropy measure value was normalised between 0 and 1. In this case, 1 represents high diversity while 0 represents no diversity.

To calculate entropy, values of each dimension of particle positions are first placed into a number of bins $b$. The $b$-ary entropy measure of the probability distribution is then calculated as

$$E = - \sum_{w=1}^{b} p_w \log_b p_w \tag{7}$$

where $b$ refers to the number of bins that the values of particle positions have been binned into. $p_w$ refers to the probability of a value of a one-dimensional, particle position to be in a bin $w$; that is, the number of particles in bin $w$ divided by the total number of particles used in the search space.

To adapt Equation (7) to multi-dimensional problems, each dimension $d$ of particle positions is first placed into $b$ bins. The entropy for each dimension $d$ over all particle positions in that dimension is then calculated. The average is then calculated from the ensuing $D$ entropy measures as

$$D_{\hat{\text{E}}} = \frac{1}{D} \sum_{d=1}^{D} E_d \tag{8}$$

where $D$ is the number of dimensions for the problem in question. The entropy of each dimension $d$ is computed as $E_d = -\sum_{w=1}^{b} p_{w,d} \log_b p_{w,d}$, where $p_{w,d}$ is the probability of a particle position in dimension $d$ to be in $w$.

The value derived from the entropy measure quantifies the uncertainty of drawing a solution from the swarm of candidate solutions. However, it does not indicate whether there are any solutions converged at certain locations of the objective space. Since the goal of niching algorithms is to locate niches of solutions for different optima, the entropy measure needs to be adapted for niching algorithms. In the work reported here, the entropy, $D_{\hat{\text{E}}}$, was calculated per niche, then averaged.

### 2.1.6. Solow–Polasky Diversity

Preuss and Wessing [4] proposed the use of the Solow–Polasky diversity measure (SPD) [18] as a technique to quantify diversity in niching algorithms. The SPD is computed by first creating an $n \times n$ matrix, $M$, whose entries make use of the candidate solutions' positions. The entries of the matrix are defined as $m_{i,j} = \exp(-\theta\|\mathbf{x}_i - \mathbf{x}_j\|_2)$. Consequently, $M$ is a correlation matrix between candidate solutions $\mathbf{x}_i$ and $\mathbf{x}_j$ [18]. The parameter $\theta$ is a user defined variable that normalises the relationship between the distance $\|\mathbf{x}_i - \mathbf{x}_j\|_2$ and $n$, which is the total number of candidate solutions [9]. Preuss and Wessing [4] have shown that the choice of the value of $\theta$ is problem dependent, and therefore SPD is only efficient if $\theta$ is tuned per problem.

Once the matrix is constructed, the SPD value is then calculated as

$$D_{\text{SPD}} = e^\top M^{-1} e \tag{9}$$

where $e = (1, ..., 1)^\top$ and $e^\top$ is its transpose.

From Equation (9), it can be deduced that $D_{\text{SPD}}$ is the sum of all entries of $M^{-1}$.

As shown by Equation (9), the SPD measure relies on the use of the inverse of the matrix, $M$. This means that, in circumstances where the matrix is singular, an inverse cannot be taken, rendering the diversity measure unusable.

### 2.1.7. Swarm Diameter and Swarm Radius

Olorunda and Engelbrecht [11] investigated the use of swarm diameter (SDM) and swarm radius (SR) in quantifying dispersion. These two measures are calculated with respect to the swarm, where the SDM is the maximum distance between any two candidate solutions. SDM is calculated as

$$D_{\text{SDM}} = \max_{\substack{\mathbf{x}_i \\ \mathbf{x}_j \in S \\ j \neq i}} \left\| \mathbf{x}_i - \mathbf{x}_j \right\|_2 \tag{10}$$

The SR, on the other hand, is the maximum distance between the centre of the swarm (or the candidate with the best objective function value) and any of the candidate solutions. SR is calculated as

$$D_{\text{SR}} = \max_{\substack{\mathbf{x}_i \in S \\ i = 1}} \left\| \mathbf{x}_i - \bar{\mathbf{x}} \right\|_2 \tag{11}$$

When utilised for measuring diversity for non-niching techniques, Olorunda and Engelbrecht [11] showed that SDM and SR are greatly affected by the presence of outliers. Nevertheless, these measures could be modified to compute diversity in niching algorithms by subjecting each candidate niche to the measures and then calculating the mean. For instance, a modified SDM is calculated as

$$D_{m\text{SDM}} = \frac{1}{\mu} \sum_{k=1}^{\mu} \max_{\substack{\mathbf{x}_i \\ \mathbf{x}_j \in N_k \\ j \neq i}} \left\| \mathbf{x}_i - \mathbf{x}_j \right\|_2 \tag{12}$$

### 2.1.8. Other Measures of Diversity

Olorunda and Engelbrecht [11] also studied the normalised average distance around the swarm center and the swarm coherence. The normalised average distance around the swarm center is similar to ADSC with the only exception that a further normalisation with regard to the diameter of the swarm is carried out. The work in [11] showed that the normalised average distance around the swarm center followed a uniform distribution between [0,1] and was thus rendered unsuitable to measure diversity over time. Whereas the goal of this work is not to measure diversity over the number iterations, since the

diversity of the solutions of niching algorithms is affected by the extent of the exploration carried out by the swarm, average distance around the swarm center is unlikely to work where multiple optima are considered. As such, it is not considered in this study.

Furthermore, Olorunda and Engelbrecht [11] showed that the results of swarm coherence were unequivocal and hence not suitable for measuring diversity for non-niching techniques. Since swarm coherence is based on average speed of particles in a swarm relative to that of the swarm centre, this measure is applicable to only PSO techniques. As such, its applicability to other complementary niching techniques cannot be determined.

### 2.2. Niche Diversity

At the beginning of a search process, candidate solutions are scattered around the search space. Depending on how a niching algorithm defines a niche, all particles at the beginning of a search process may be categorised as belonging to one niche, i.e., the entire swarm is referred to as a niche. Alternatively, each particle could represent its own niche. That is, if a swarm has $n$ particles, then there exist $n$ niches. As iterations increase, particles start to group together, forming candidate niches. The grouping could be based on an already defined niche radius or a particular algorithm may have its own niche determination strategy. Each candidate niche contains a neighbourhood best (i.e, the best candidate in a niche also known as the niche best. This does not mean that a neighbourhood topology is being used.) or a candidate solution that has the best fitness within that niche.

Niche diversity refers to diversity with respect to the neighbourhood bests in each of the identified candidate niches. For instance, in Figure 1b, there are four identified candidate niches clustered around the position of the optima. To calculate niche diversity using any of the methods discussed in Section 2.1, only the four neighbourhood bests will be considered.

A niching algorithm will have many candidate niches during exploration. These candidate niches are later merged if it is determined that they are converging towards the same optimum. Different niching algorithms have different merging strategies. However, merging occurs during the exploitation phase. Due to this, niche diversity is expected to be high during the exploration phase and starts to decrease as the exploitation phase starts. However, at the end of the search process, niche diversity is still expected to be high if many of the candidate niches were located and maintained. Algorithms that are not able to maintain found potential niches will have a low diversity at the end of the search process, that is, all candidate niches have converged towards one optimum.

Unlike swarm diversity, niche diversity helps to determine how candidate niches identify and maintain niches. A good niching algorithm is expected to have a considerable high diversity both at the exploration and exploitation phases of the search process, i.e., for a good algorithm, niche diversity is not expected to be zero unless only one optimum exists in the search space. However, niche diversity is still affected by such drawbacks as "outliers" and niche identification strategies.

### 3. Materials and Methods

In this section, the methodology used to conduct the empirical analysis is discussed. Furthermore, a method to correctly determine unique solutions is introduced.

The main objective of the empirical analysis conducted for this paper is to illustrate the applicability of the discussed measures to correctly reflect diversity for niching algorithms. As discussed earlier, diversity in niching algorithms can be calculated with respect to the decision space (i.e., swarm diversity) and also with respect to solution space (i.e., niche diversity).

In order to calculate the entropy measure, the values of each dimension of particle positions were divided into different numbers of bins. The number of bins used were 10, 20, 50 and 100 in each dimension. The number of bins were used as the logarithm base for the entropy measure defined in Equation (8). There is, therefore, four types of entropies

that were carried out, i.e., $\hat{E}_{10}$, $\hat{E}_{20}$, etc. The reason for using different numbers of bins is to determine how the number of bins affect diversity values.

The iterated F-Race algorithm [19] was used to tune the ESPSO parameters. Parameter tuning was carried out for every problem listed in Table 1. Consequently, different parameters were obtained and are therefore not listed in this paper. The ESPSO algorithm was implemented using CIlib (http://www.cilib.net).

**Table 1.** Multimodal functions used for the experiments.

| Function & Name | Characteristics |
|---|---|
| Inverted Branin's RCOS<br>$\mathbf{f}_1 = -((x_2 - \frac{5.1x_1^2}{4\pi^2} + \frac{5x_1}{\pi} - 6)^2 + 10(1 - \frac{1}{8\pi})\cos x_1 + 10)$ | 3 global peaks<br>$x_1 \in [-5, 10]$<br>$x_2 \in [0, 15]$ |
| Inverted Egg Holder<br>$\mathbf{f}_2 = -\sum_{i=1}^{n-1}(-(x_{i+1} + 47)\alpha + \beta(-x_i))$<br>$\alpha = \sin(\sqrt{|x_{i+1} + x_i/2 + 47|})$<br>$\beta = \sin(\sqrt{|x_i - (x_{i+1} + 47)|})$ | 1 global peak<br>Many local peaks<br>$\mathbf{x}_i \in [-512, 512]^2$ |
| Equal Maxima<br>$\mathbf{f}_3 = \sum_{i=1}^{n}\sin^6(5\pi x_i)$ | $5^n$ global peaks<br>$\mathbf{x}_i \in [0, 1]^2$ |
| Modified Himmelblau<br>$\mathbf{f}_4 = 200 - (x_1^2 + x_2 - 11)^2 + (x_1 + x_2^2 - 7)^2$ | 4 global peaks<br>$\mathbf{x}_i \in [-6, 6]^2$ |
| Inverted Michalewicz<br>$\mathbf{f}_5 = \sum_{i=1}^{n}[\sin(x_i).\sin^{20}(\frac{i*x_i^2}{\pi})]$ | 1 global peak<br>$n! - 1$ local peaks<br>$\mathbf{x}_i \in [0, \pi]^2$ |
| Inverted Rastrigin<br>$\mathbf{f}_6 = -\sum_{i=1}^{n}[x_i^2 - 10\cos(2\pi x_i) + 10]$ | 1 global peak<br>Many local peaks<br>$\mathbf{x}_i \in [0, \pi]^2$ |
| Inverted Rosenbrock<br>$\mathbf{f}_7(\vec{x}) = -\sum_{i=1}^{n-1}\{(100(x_{i+1} - x_i^2)^2$<br>$+ (x_i - 1)^2)\}$ | 1 global peak<br>Many local peaks<br>$\mathbf{x}_i \in [-5, 5]^4$ |
| Inverted Schwefel Problem 2_26<br>$\mathbf{f}_8 = -\sum_{i=1}^{n} x_i \sin(\sqrt{|x_i|})$ | 1 global peaks<br>$8^n - 1$ local peaks<br>$\mathbf{x}_i \in [-500, 500]^2$ |
| Inverted Six-Hump Camel Back<br>$\mathbf{f}_9 = \sum_{i=1}^{n-1}\{(4 - 2.1x_i^2 + \frac{x_i^4}{3})x_i^2 + x_i x_{i+1} + (-4 + 4x_{i+1}^2)x_{i+1}^2\}$ | 2 global peaks<br>4 local peaks<br>$\mathbf{x}_i \in [0, \pi]^2$ |

To ensure that the obtained results were not affected by randomisation, 30 independent runs were carried out for every function listed in Table 1. For each measure, the same starting points were used. The reader is directed to Appendices D and E for a visual representation of these functions in two dimensions. Each epoch was made up of 1000 iterations. The population size for the ESPSO algorithm was made up of 100 particles. During the search, particles that bounced off the search space were randomly re-initialised within the search space. The Inverted Rosenbrock function ($f_7$), was investigated in four dimensions while the rest of the functions were tested in two-dimensional search spaces. The results obtained from the 30 independent runs were used to calculate all the investigated diversity measures using the following equation

$$z' = \frac{1}{30}\sum_{i}^{30} z_i \tag{13}$$

where $z'$ is the scaled (reported) diversity value, $z_i$ is the observed/computed diversity value for a particular diversity measure in the $i$th independent run.

*Determining Unique Solutions*

At the end of a niching algorithm run, the best solution in each candidate niche is usually reported. It is possible for an algorithm run to terminate before all optima have been found and/or before particles optimising a particular location of optimum fully converged. In order to correctly report the obtained results, it is important to evaluate the reported solutions in order to weed out candidate solutions that were optimising the same optimum. This helps to remove duplicate solutions and to report only those solutions that are unique in terms of their positions in the search space.

In order to determine the unique solutions to be used for the measurement of the niche diversity, the work reported here used the mid-point technique [20]. The mid-point technique ascertains that a particular candidate solution, $\check{x}_i$, is distinctive from another candidate solution, $\check{x}_j$, by determining whether the relation shown by Equation (14) holds (assuming maximisation).

$$f(\bar{\mathbf{x}}_k) \geq \min\{f(\check{x}_i), f(\check{x}_j)\} \mid \forall k \in \{1, ..., m\} \tag{14}$$

where $\bar{\mathbf{x}}_k$ is a point between $\check{x}_i$ and $\check{x}_j$ and $m$ is the total number of points between $\check{x}_i$ and $\check{x}_j$. If the relation does not hold, then a local minimum exists between $\check{x}_i$ and $\check{x}_j$, implying that both are moving towards different maxima; i.e., $\check{x}_i$ and $\check{x}_j$ represent different niches. If the relation holds, then the candidate solution with the better objective function value is considered as the unique solution. Note that $m = 10$ is used in the experiments here as it was shown to be effective in [20]. In addition, the $m$ points were sampled at equal intervals between $\check{x}_i$ and $\check{x}_j$.

## 4. Results and Discussion

The purpose of this section is to present and discuss the results of the observed diversity values as quantified using the previously discussed diversity measures. Since the goal is to review and analyse the effectiveness of each of the listed measures in quantifying diversity; for the remainder of this section, each diversity measure is evaluated on its performance on all the listed multimodal problems and a conclusion is drawn. The curves shown by the figures are values of each diversity measure as an average of the 30 independent runs over 1000 iterations. There is no attempt made at comparing the measures directly.

### 4.1. Sum of Distances

Table 2 as well as Figures 3 and 4 show the quantification of diversity using the sum of distance measure (SD) and its proposed variants, i.e., modified SD (mSD) and niche SD (nSD).

**Table 2.** Mean ($\mu$) and standard deviation ($\sigma$) for SD measure at the last iteration.

| Function | Mean (SD) | Stdev (SD) | Mean (mSD) | Stdev (mSD) | Mean (nSD) | Stdev (nSD) |
|---|---|---|---|---|---|---|
| $f_1$ | $3.02 \times 10^2$ | $9.12 \times 10^0$ | $2.76 \times 10^1$ | $1.01 \times 10^1$ | $1.41 \times 10^1$ | $4.40 \times 10^0$ |
| $f_2$ | $3.62 \times 10^3$ | $4.87 \times 10^2$ | $7.93 \times 10^0$ | $8.11 \times 10^0$ | $1.31 \times 10^3$ | $4.47 \times 10^2$ |
| $f_3$ | $7.39 \times 10^1$ | $2.30 \times 10^0$ | $1.13 \times 10^0$ | $2.67 \times 10^{-1}$ | $2.15 \times 10^1$ | $2.57 \times 10^0$ |
| $f_4$ | $2.26 \times 10^2$ | $5.43 \times 10^0$ | $1.38 \times 10^1$ | $2.74 \times 10^0$ | $1.31 \times 10^1$ | $2.17 \times 10^0$ |
| $f_5$ | $3.32 \times 10^1$ | $2.50 \times 10^1$ | $0.00 \times 10^0$ | $0.00 \times 10^0$ | $3.32 \times 10^1$ | $2.50 \times 10^1$ |
| $f_6$ | $9.22 \times 10^1$ | $1.82 \times 10^1$ | $2.14 \times 10^{-1}$ | $3.37 \times 10^{-1}$ | $1.94 \times 10^1$ | $1.14 \times 10^1$ |
| $f_7$ | $4.30 \times 10^2$ | $5.75 \times 10^1$ | $4.28 \times 10^1$ | $2.19 \times 10^1$ | $1.70 \times 10^1$ | $8.44 \times 10^0$ |
| $f_8$ | $3.35 \times 10^3$ | $4.11 \times 10^2$ | $2.03 \times 10^{-1}$ | $2.03 \times 10^{-1}$ | $3.32 \times 10^3$ | $4.15 \times 10^2$ |
| $f_9$ | $1.41 \times 10^2$ | $1.15 \times 10^1$ | $8.32 \times 10^0$ | $3.47 \times 10^0$ | $1.09 \times 10^1$ | $3.45 \times 10^0$ |

The results for all the considered multimodal problems show that the diversity quantified using the standard SD is high. For swarm diversity, the expectation is for diversity to decrease monotonically as the particles converge towards the locations of optima. However, for the standard SD, since the pairwise distance of each particle to all other particles in the swarm is considered, the diversity predicted will be high. These results show that the standard SD is not suitable for niching algorithms.

For the mSD, the expectation is that, as the particles converge towards the location of optima, the measured diversity should monotonically decrease. It is also expected that, if each particle forms a niche, then the diversity predicted should be low. The results obtained correlates to the expectations, that is, the obtained mSD predicts low diversity towards convergence.

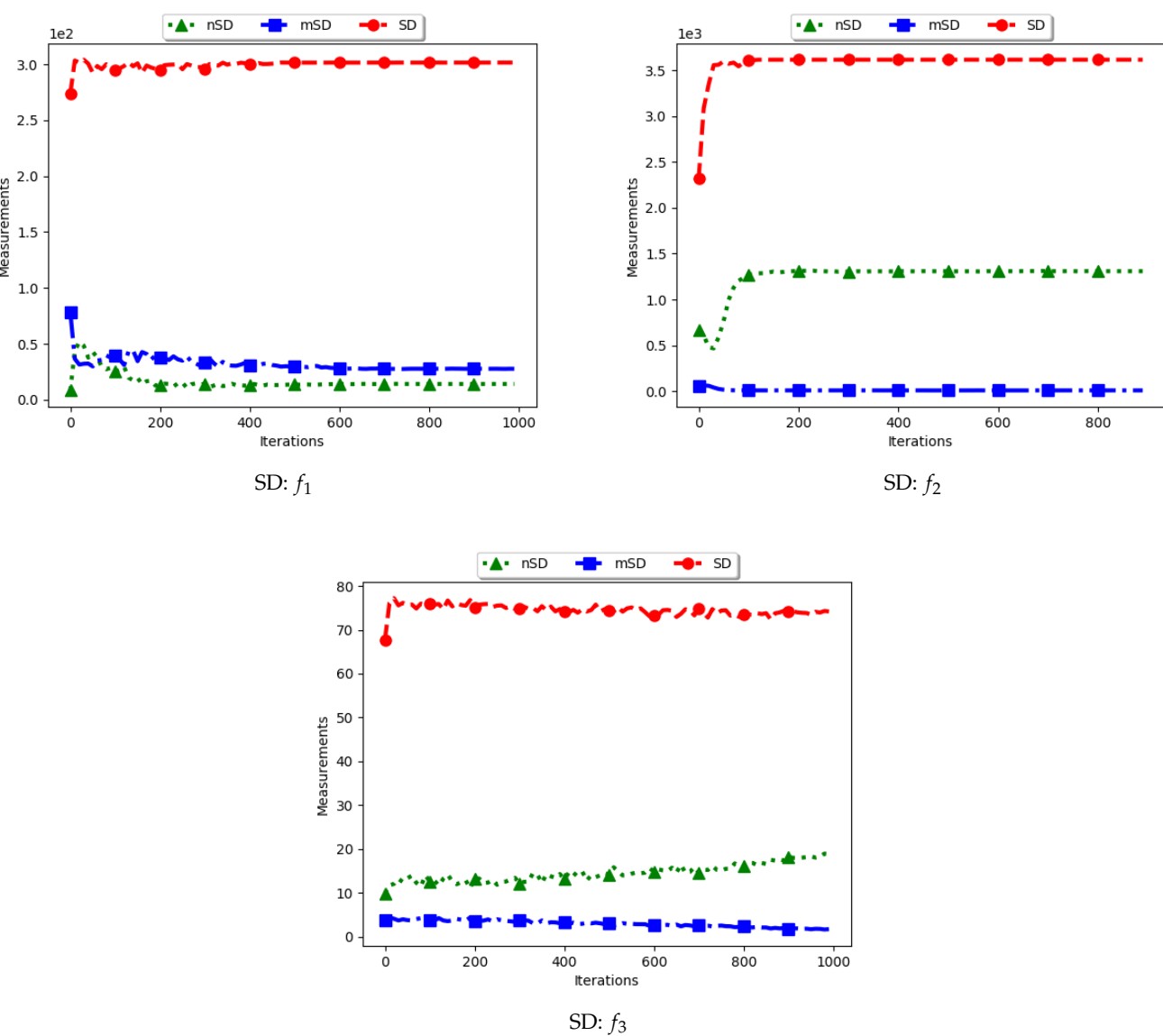

**Figure 3.** Quantification of diversity using the sum of distance measure.

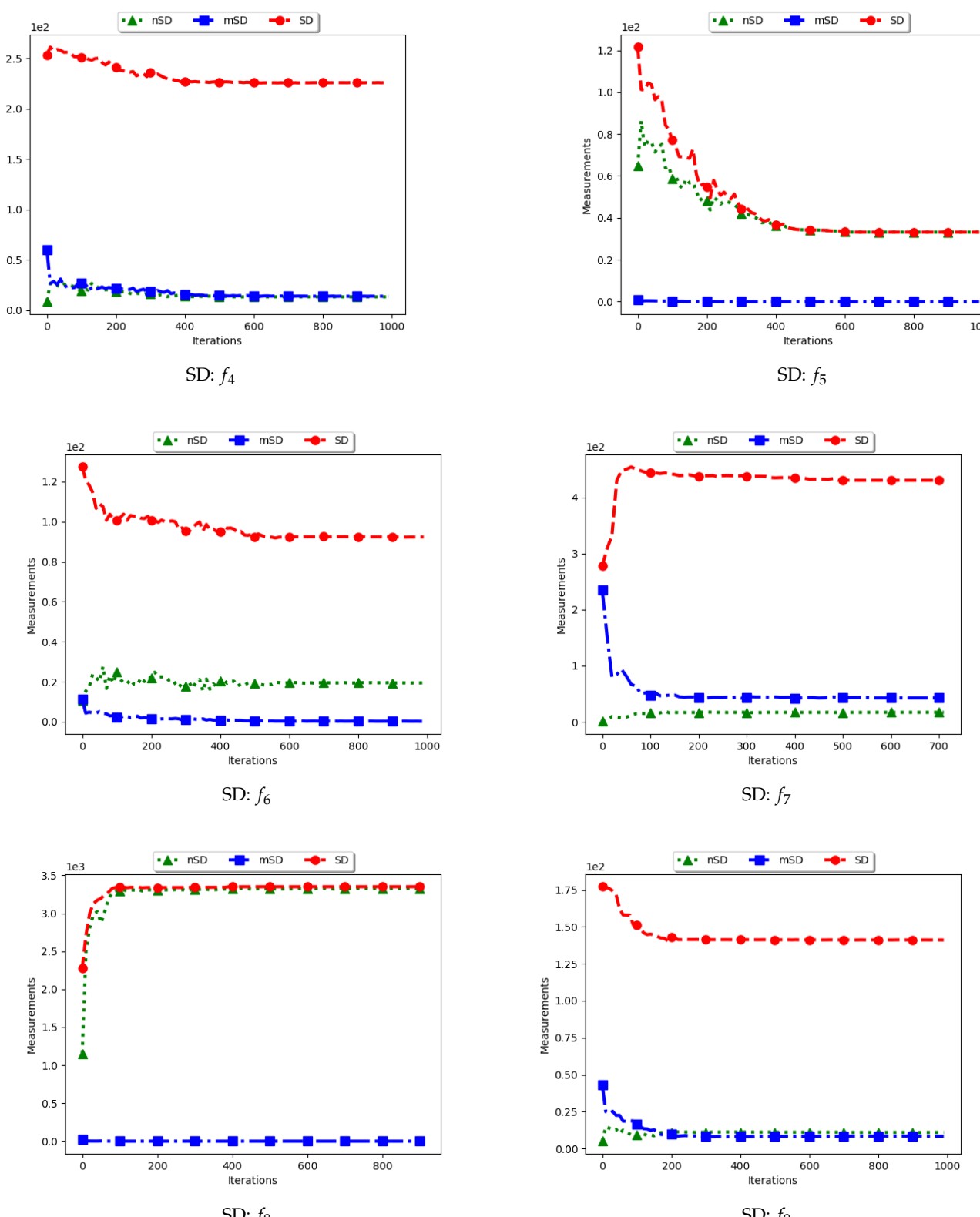

**Figure 4.** Quantification of diversity using the sum of distances measure (*Cont.*).

The nSD calculates diversity as standard SD does, but only using the unique solutions. As such, achieved diversity should be high. In addition, even in the case where the unique solutions are close to each other, it is expected that the diversity will never converge. The obtained results agree with this expectation.

In conclusion, the standard SD is not a good measure to quantify diversity in niching algorithms, as is expected. If the diversity of all candidate solutions is required, the mSD should be used. Finally, if only the diversity of found solutions is required, then nSD is a good measure.

### 4.2. Sum of Distances to the Nearest Neighbour

Figures 5 and 6 and Table 3 show the quantification of diversity using the sum of distance to the nearest neighbour measure (SDNN) and its proposed variants, i.e., modified SDNN (mSDNN) and niche SDNN (nSDNN).

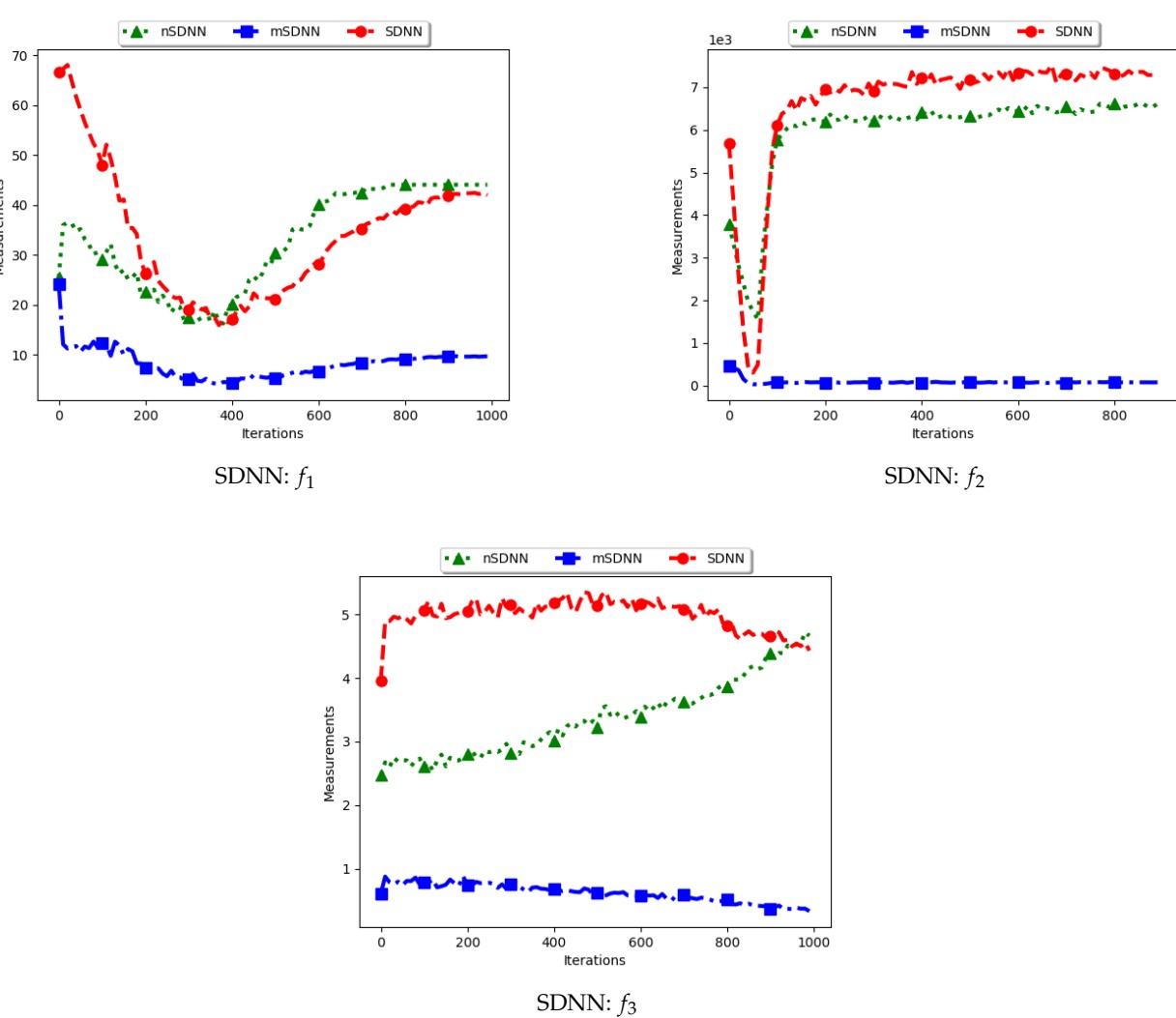

**Figure 5.** Quantification of diversity using the sum of distances to the nearest neighbour.

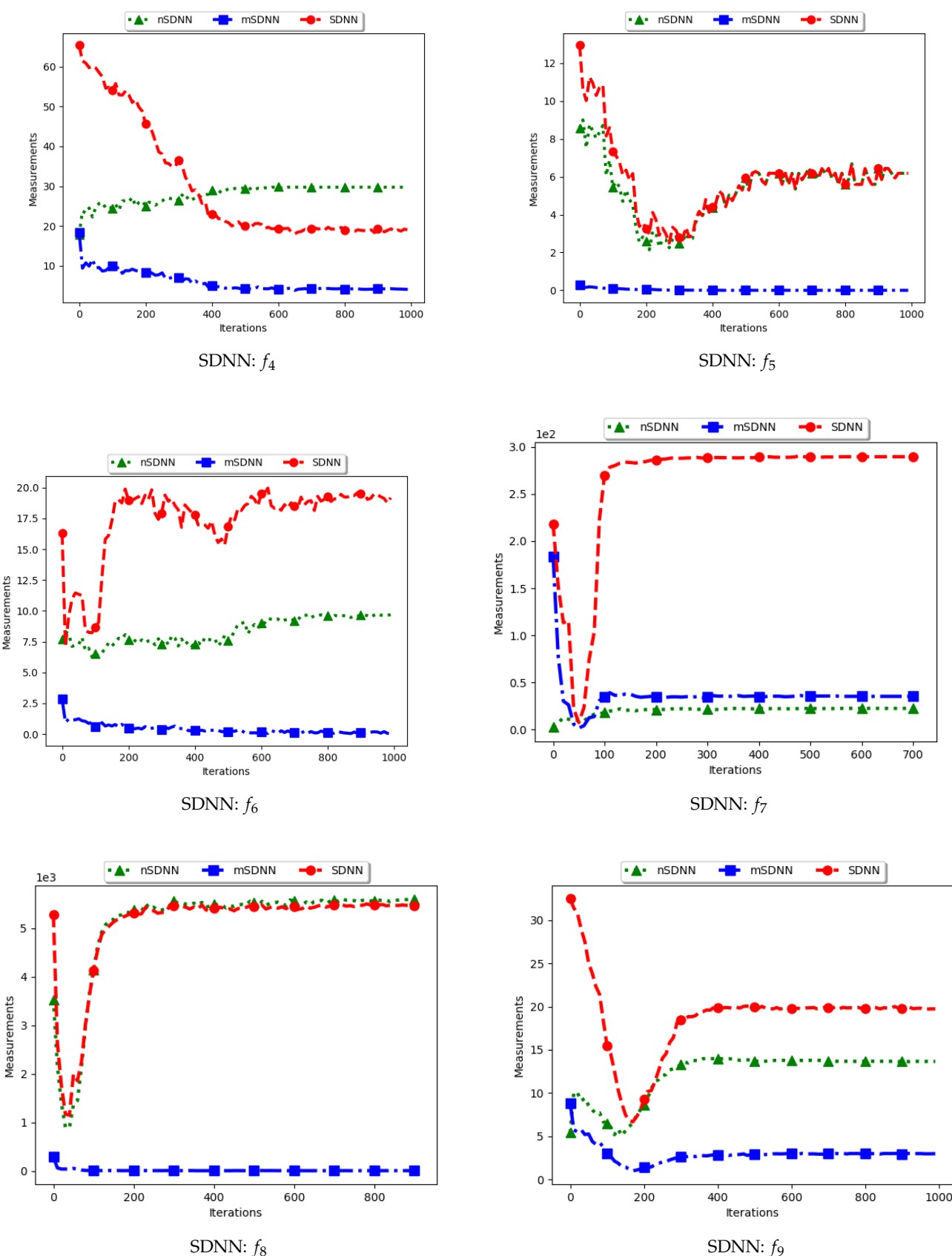

**Figure 6.** Quantification of diversity using the sum of distances to the nearest neighbour (*Cont.*).

**Table 3.** Mean ($\mu$) and standard deviation ($\sigma$) for SDNN measure at the last iteration.

| Function | Mean (SDNN) | Stdev (SDNN) | Mean (mSDNN) | Stdev (mSDNN) | Mean (nSDNN) | Stdev (nSDNN) |
|---|---|---|---|---|---|---|
| $f_1$ | $4.28 \times 10^1$ | $1.29 \times 10^1$ | $9.77 \times 10^0$ | $4.09 \times 10^0$ | $4.41 \times 10^1$ | $1.41 \times 10^1$ |
| $f_2$ | $7.29 \times 10^3$ | $3.03 \times 10^3$ | $7.52 \times 10^1$ | $1.13 \times 10^2$ | $6.54 \times 10^3$ | $2.37 \times 10^3$ |
| $f_3$ | $4.26 \times 10^0$ | $5.30 \times 10^{-1}$ | $2.35 \times 10^{-1}$ | $5.02 \times 10^{-2}$ | $5.58 \times 10^0$ | $7.37 \times 10^{-1}$ |
| $f_4$ | $1.89 \times 10^1$ | $3.73 \times 10^0$ | $4.22 \times 10^0$ | $1.39 \times 10^0$ | $2.97 \times 10^1$ | $5.04 \times 10^0$ |
| $f_5$ | $6.18 \times 10^0$ | $6.61 \times 10^0$ | $0.00 \times 10^0$ | $0.00 \times 10^0$ | $6.18 \times 10^0$ | $6.61 \times 10^0$ |
| $f_6$ | $1.88 \times 10^1$ | $1.78 \times 10^1$ | $1.52 \times 10^{-1}$ | $4.77 \times 10^{-1}$ | $9.62 \times 10^0$ | $3.96 \times 10^0$ |
| $f_7$ | $2.90 \times 10^2$ | $4.15 \times 10^1$ | $3.51 \times 10^1$ | $1.91 \times 10^1$ | $2.23 \times 10^1$ | $9.92 \times 10^0$ |
| $f_8$ | $5.47 \times 10^3$ | $1.67 \times 10^3$ | $6.40 \times 10^0$ | $7.23 \times 10^0$ | $5.59 \times 10^3$ | $1.63 \times 10^3$ |
| $f_9$ | $1.97 \times 10^1$ | $7.05 \times 10^0$ | $2.97 \times 10^0$ | $1.70 \times 10^0$ | $1.37 \times 10^1$ | $4.03 \times 10^0$ |

The standard SDNN obtains diversity as the sum of all the nearest neighbours for the particles in the swarm. As such, it is expected that, as the particles converge towards the optima, SDNN will obtain low diversity scores because a nearest neighbour of a particle will be another particle in the same niche. The results showed in Figures 5 and 6 do not seem to correlate with the expected behaviour. The figures show that diversity starts to decrease monotonically and then rises either at the 100th iteration ($f_2, f_6, f_7, f_8$) or around the 400th iteration ($f_1, f_5$) iteration. A trace of the particle positions at various iterations (refer to Appendices A–C) shows that, whereas the particles start to converge, there is a high presence of outliers (the assumption is that since these problems have 3, 1, and 1 global peaks, respectively, the presence of multiple unique solutions means that these are outliers or unknown local optima). For these outliers, the nearest neighbours are likely to be particles in already converged niches. As such, the consideration of these outliers in the calculation raises the diversity.

The mSDNN considers the nearest neighbours at the niche level, and is therefore not affected by the presence of outliers. In the discussed work, a niche could be formed using one particle and therefore the diversity at that niche is zero. As such, the results obtained mirror what is expected. This means that, although the mSDNN seems to be a good measure, the decrease of diversity does not indicate the true picture of the search space. More investigations into this are required.

For the nSDNN, the calculation is based on the unique solutions found. Since the outliers will be considered as unique solutions, this has the effect of raising the niche diversity. As such, although the expectation is for the nSDNN to show high diversity in both the exploration and exploitation phases, outliers do play a role in the value obtained.

In summary, the SDNN as a measure has shown to be highly affected by the presence of outliers. As such, although the results obtained by the nSDNN is as expected, if this measure is used for quantifying dispersion for candidate solutions of niching algorithms, care has to be taken and a technique devised to remove the outliers.

### 4.3. Average Distance around the Swarm Centre

Figures 7 and 8 and Table 4 show the quantification of diversity using the average distance around the swarm centre measure (ADSC) and its proposed variants, i.e., modified ADSC (mADSC) and niche ADSC (nADSC). The intial posit in the article was that standard ADSC and the nASDC will show high diversity because the distances will be computed from a swarm "best" in both cases. The results do not seem to correlate with this expectation. Analysis of the ADSC led to the following observations.

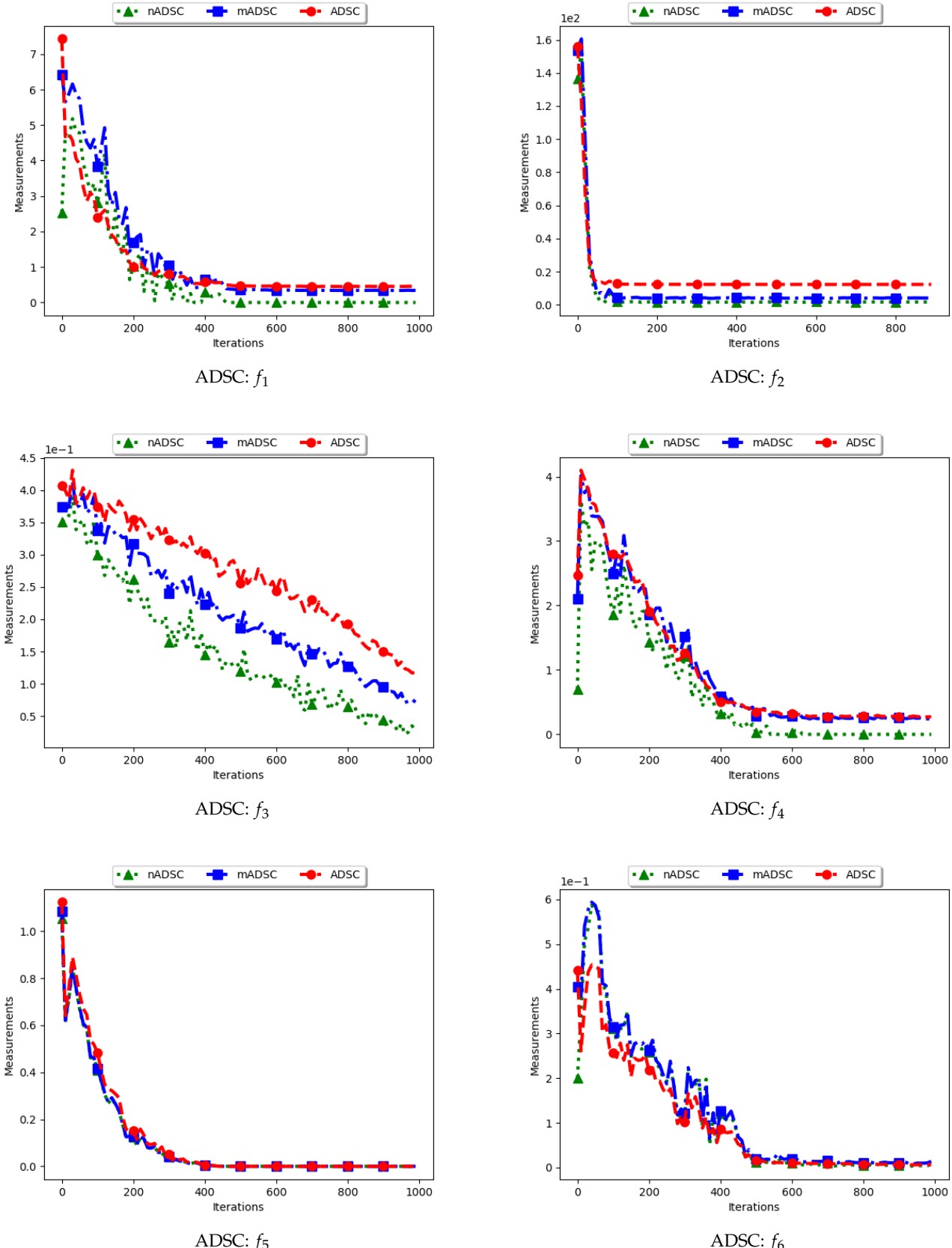

**Figure 7.** Quantification of diversity using the average distance around the swarm centre measure.

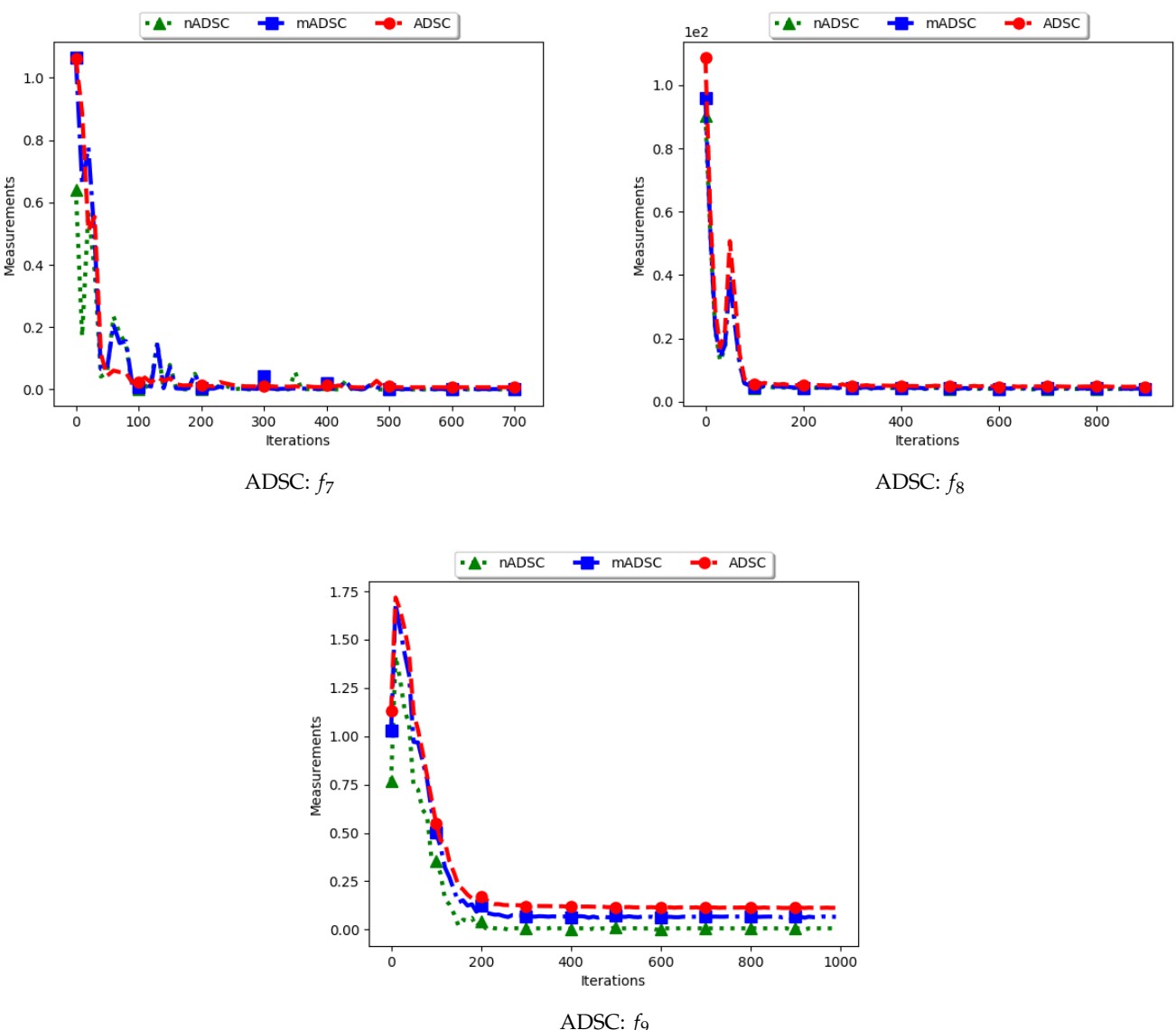

ADSC: $f_7$

ADSC: $f_8$

ADSC: $f_9$

**Figure 8.** Quantification of diversity using the average distance around the swarm centre measure (*Cont.*).

**Table 4.** Mean ($\mu$) and standard deviation ($\sigma$) for ADSC measure and its variants at the last iteration.

| Function | Mean (ADSC) | Stdev (ADSC) | Mean (mADSC) | Stdev (mADSC) | Mean (nADSC) | Stdev (nADSC) |
|---|---|---|---|---|---|---|
| $f_1$ | $4.55 \times 10^{-1}$ | $1.62 \times 10^{-1}$ | $3.42 \times 10^{-1}$ | $1.54 \times 10^{-1}$ | $0.00 \times 10^0$ | $0.00 \times 10^0$ |
| $f_2$ | $1.23 \times 10^1$ | $9.74 \times 10^0$ | $4.06 \times 10^0$ | $3.54 \times 10^0$ | $1.65 \times 10^0$ | $2.59 \times 10^0$ |
| $f_3$ | $6.54 \times 10^{-2}$ | $1.23 \times 10^{-2}$ | $3.52 \times 10^{-2}$ | $7.87 \times 10^{-3}$ | $5.95 \times 10^{-3}$ | $6.99 \times 10^{-3}$ |
| $f_4$ | $2.69 \times 10^{-1}$ | $6.93 \times 10^{-2}$ | $2.44 \times 10^{-1}$ | $1.32 \times 10^{-1}$ | $0.00 \times 10^0$ | $0.00 \times 10^0$ |
| $f_5$ | $8.45 \times 10^{-10}$ | $3.74 \times 10^{-9}$ | $8.45 \times 10^{-10}$ | $3.74 \times 10^{-9}$ | $8.45 \times 10^{-10}$ | $3.74 \times 10^{-9}$ |
| $f_6$ | $5.27 \times 10^{-3}$ | $9.36 \times 10^{-3}$ | $7.67 \times 10^{-3}$ | $1.21 \times 10^{-2}$ | $1.42 \times 10^{-3}$ | $5.35 \times 10^{-3}$ |
| $f_7$ | $6.81 \times 10^{-3}$ | $1.10 \times 10^{-2}$ | $1.03 \times 10^{-3}$ | $1.83 \times 10^{-3}$ | $1.41 \times 10^{-6}$ | $7.58 \times 10^{-6}$ |
| $f_8$ | $4.76 \times 10^0$ | $3.44 \times 10^0$ | $4.15 \times 10^0$ | $2.71 \times 10^0$ | $3.98 \times 10^0$ | $2.71 \times 10^0$ |
| $f_9$ | $1.13 \times 10^{-1}$ | $4.00 \times 10^{-2}$ | $6.54 \times 10^{-2}$ | $4.15 \times 10^{-2}$ | $6.34 \times 10^{-3}$ | $3.42 \times 10^{-2}$ |

The standard ADSC sums the distances from each particle to the best particle in the swarm and then computes the average. For niching algorithms, there is a high likelihood of multiple global bests, based on the modality of the problem being investigated. As such, each particle's distance from the global best is dependent on which optimum the particle was optimising. This behaviour explains why the diversity as quantified using the ADSC decreased monotonically as the search progressed.

For the mADSC, it was expected that the diversity will decrease monotonically. This is because the computation is done using the best particle in the niche.

Similar to the standard ADSC, the nADSC measure was computed using the nbests. As such, although the final particle positions (as shown in Appendices A–C) suggest that, in the presence of outliers, the use of nbests in the computation meant that there were multiple "centres" being used. This led to the low diversity.

These results suggest that a better strategy needs to be devised in working with both ADSC and nADSC. Further work on this area will compare the use of a central position in the swarm with the current use of a swarm "best".

### 4.4. Average of the Mean Distance around All Solutions

Figures 9 and 10 as well as Table 5 show the quantification of diversity using the average of the mean distance around all solutions (ADAA) and its proposed variants, i.e., modified ADAA (mADAA) and niche ADAA (nADAA).

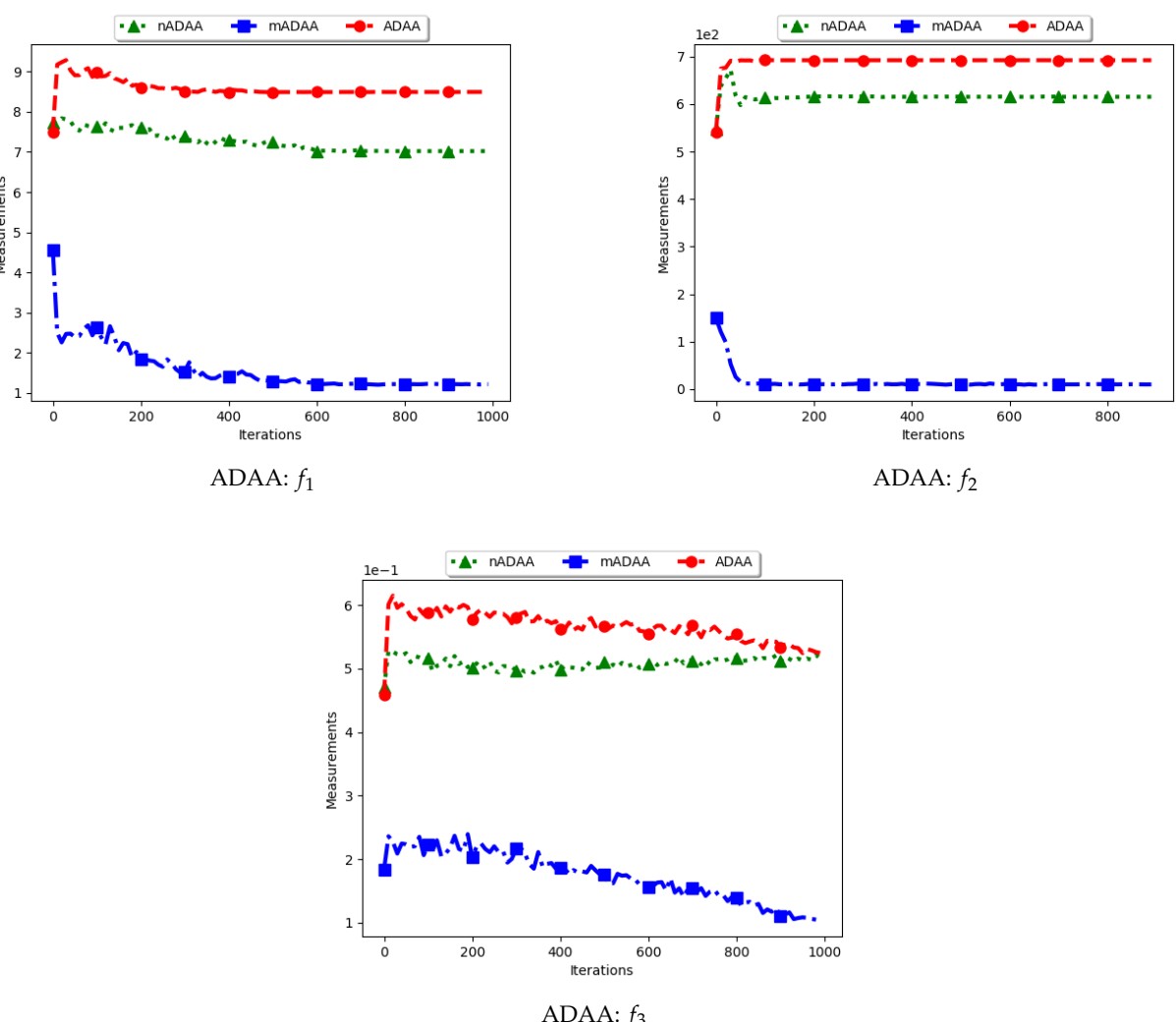

**Figure 9.** Quantification of diversity using the average of the mean distance around all solutions.

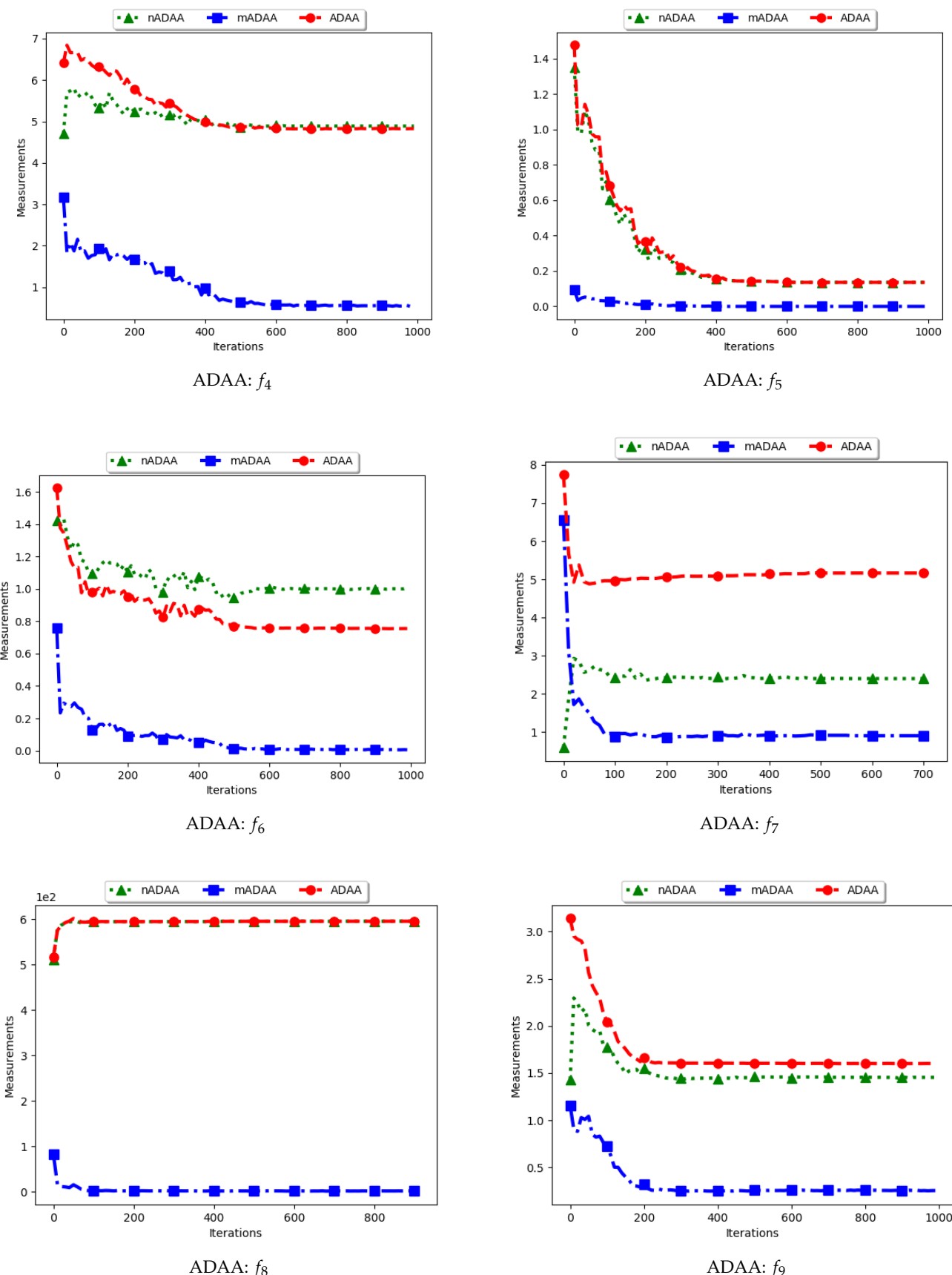

**Figure 10.** Quantification of diversity using the average of the mean distance around all solutions (*Cont.*).

**Table 5.** Mean ($\mu$) and standard deviation ($\sigma$) for ADAA measure and its variants at the last iteration.

| Function | Mean (ADAA) | Stdev (ADAA) | Mean (mADAA) | Stdev (mADAA) | Mean (nADAA) | Stdev (nADAA) |
|---|---|---|---|---|---|---|
| $f_1$ | $8.49 \times 10^0$ | $3.56 \times 10^{-1}$ | $1.21 \times 10^0$ | $4.91 \times 10^{-1}$ | $7.02 \times 10^0$ | $7.98 \times 10^{-1}$ |
| $f_2$ | $6.92 \times 10^2$ | $2.47 \times 10^1$ | $9.98 \times 10^0$ | $1.16 \times 10^1$ | $6.15 \times 10^2$ | $5.42 \times 10^1$ |
| $f_3$ | $5.06 \times 10^{-1}$ | $2.43 \times 10^{-2}$ | $6.83 \times 10^{-2}$ | $1.38 \times 10^{-2}$ | $5.17 \times 10^{-1}$ | $2.58 \times 10^{-2}$ |
| $f_4$ | $4.83 \times 10^0$ | $1.94 \times 10^{-1}$ | $5.59 \times 10^{-1}$ | $1.96 \times 10^{-1}$ | $4.89 \times 10^0$ | $1.67 \times 10^{-1}$ |
| $f_5$ | $1.36 \times 10^{-1}$ | $1.15 \times 10^{-1}$ | $0.00 \times 10^0$ | $0.00 \times 10^0$ | $1.36 \times 10^{-1}$ | $1.15 \times 10^{-1}$ |
| $f_6$ | $7.53 \times 10^{-1}$ | $2.75 \times 10^{-1}$ | $6.40 \times 10^{-3}$ | $1.38 \times 10^{-2}$ | $9.98 \times 10^{-1}$ | $2.63 \times 10^{-1}$ |
| $f_7$ | $5.16 \times 10^0$ | $3.39 \times 10^{-1}$ | $8.99 \times 10^{-1}$ | $4.54 \times 10^{-1}$ | $2.40 \times 10^0$ | $6.37 \times 10^{-1}$ |
| $f_8$ | $5.96 \times 10^2$ | $1.96 \times 10^1$ | $1.64 \times 10^0$ | $1.82 \times 10^0$ | $5.96 \times 10^2$ | $2.00 \times 10^1$ |
| $f_9$ | $1.60 \times 10^0$ | $2.02 \times 10^{-1}$ | $2.54 \times 10^{-1}$ | $1.31 \times 10^{-1}$ | $1.45 \times 10^0$ | $1.41 \times 10^{-1}$ |

The ADAA computes the sum of the average pairwise distances of each particle to all other particles in the swarm and then finds the average. In essence, the ADAA computes the average spread of the entire swarm. The effect is that, as the particles converge towards optima, the diversity will decrease and then stagnate at a relatively high diversity value.

The results obtained by the standard ADAA correlate with the expectation, i.e., for the diversity to decrease gradually. However, for some of the problems, e.g., $f_2$ and $f_8$, this was not the case. Analysis of the positions of the particles in various iterations showed that, for these two problems, convergence started to occur early in the search. As such, particles were only exploiting near their points of convergence, leading to larger distances from other particles. This too should be expected of this measure as niching algorithms are expected to explore widely over the entire search space.

For the mADAA, the results are as expected. The pairwise average distances was calculated with respect to each niche and then averaged. As such, it was expected that the small intra-niche distances would lead to small diversity.

The behaviour of the nADAA was somewhat similar to that of the standard ADAA. Since the pairwise distances are calculated using only unique solutions, it is therefore correct to posit that the quantified diversity will be high in both the exploitation and the exploration phase.

In summary, the nADAA is a suitable measure to quantify the spread of the solutions of niching algorithms, while the mADAA is a suitable measure to quantify diversity within the candidate niches.

*4.5. Entropy*

Figures 11 and 12 and Table 6 illustrate the diversity as measured using entropy with 20 bins. From the investigations carried out, no discernible difference was seen between $\hat{E}_{10}$, $\hat{E}_{20}$, $\hat{E}_{50}$ and $\hat{E}_{100}$. As such, the discussion is carried out using the swarm diversity measures, $\hat{E}_{20}$ and $m\hat{E}_{20}$, and the niche diversity, $n\hat{E}_{20}$.

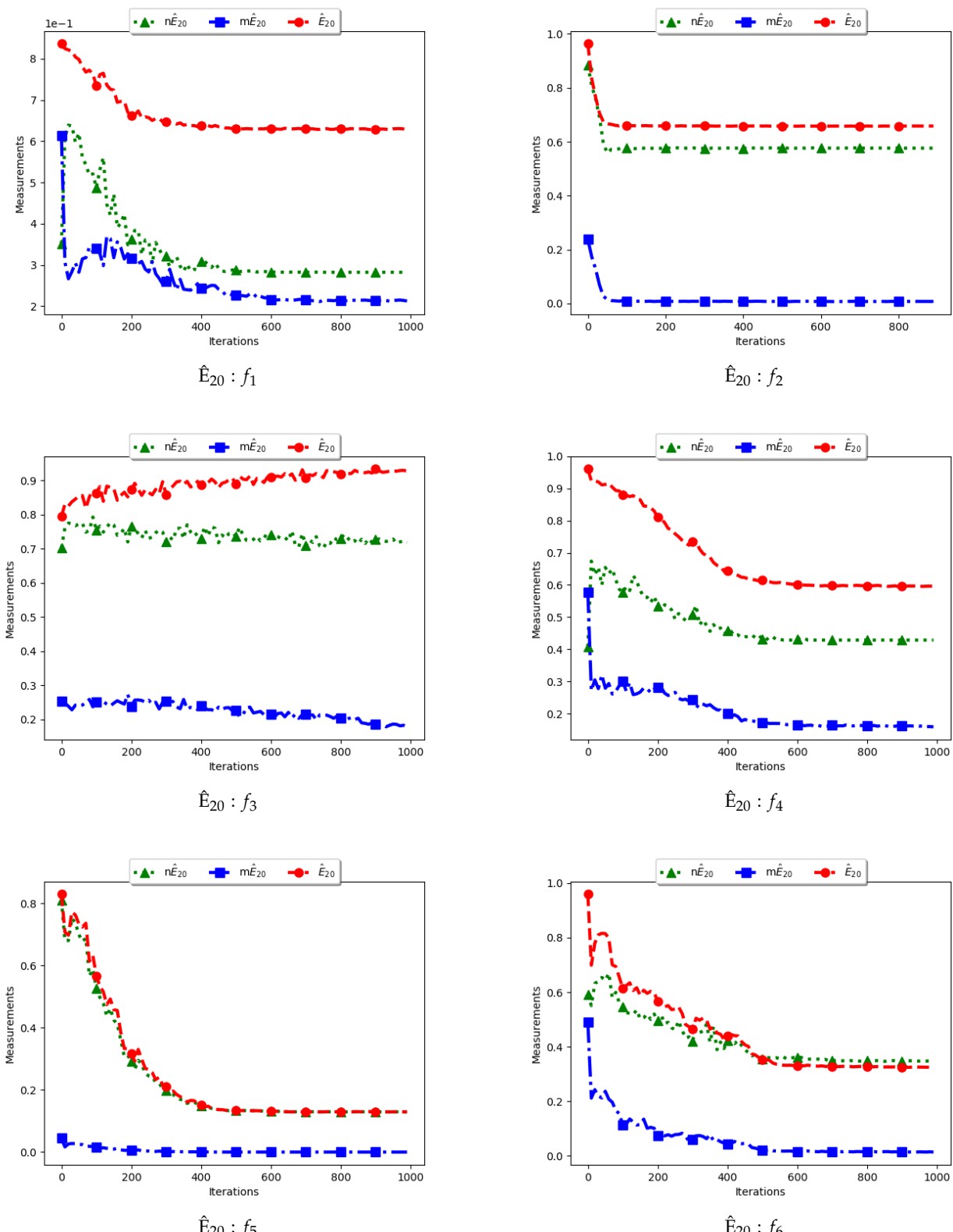

**Figure 11.** Quantification of diversity using the Entropy measure.

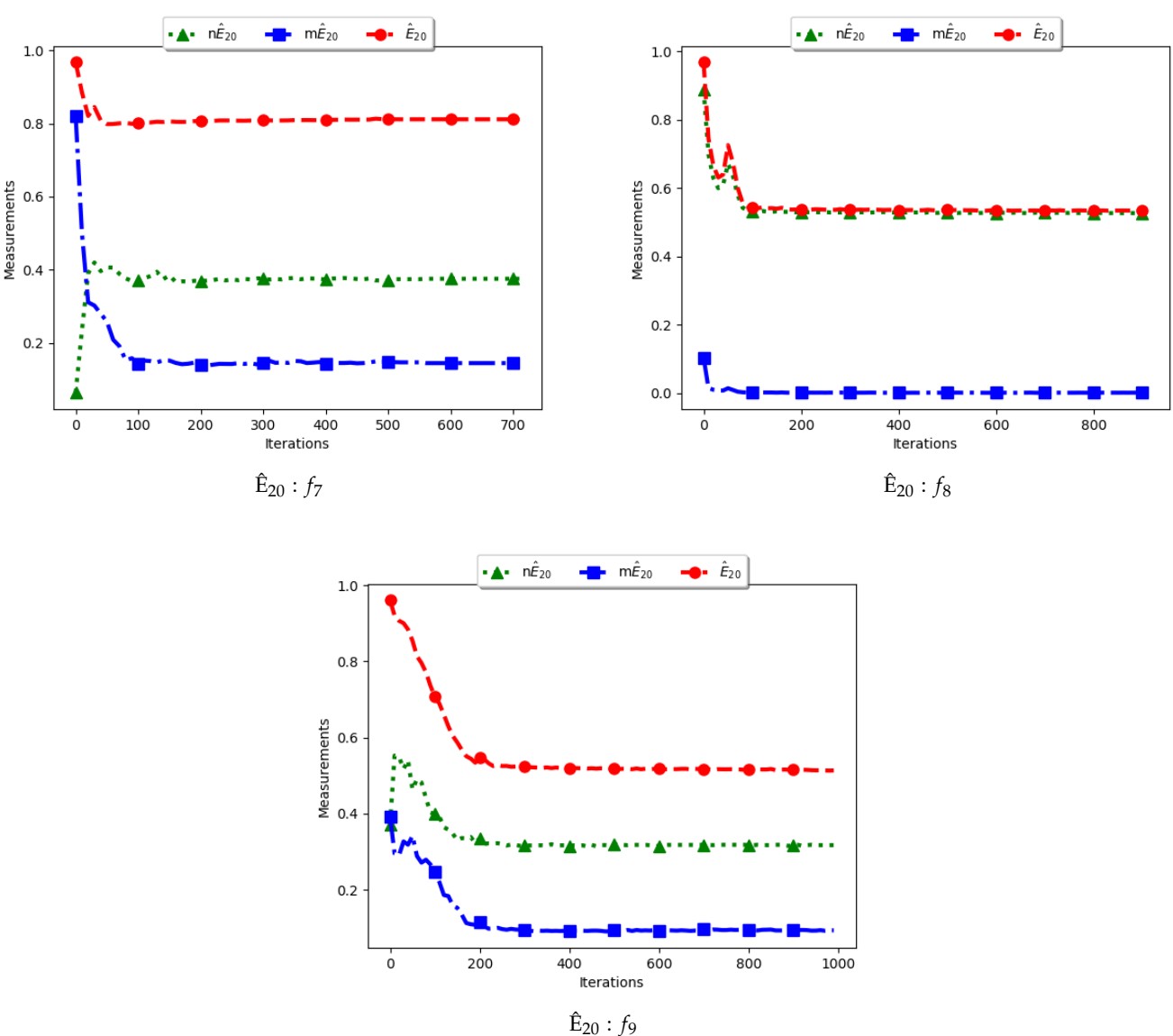

**Figure 12.** Quantification of diversity using the Entropy measure (*Cont.*).

**Table 6.** Mean ($\mu$) and standard deviation ($\sigma$) for the Entropy measure and its variants at the last iteration.

| Function | Mean ($\hat{E}_{20}$) | Stdev ($\hat{E}_{20}$) | Mean ($m\hat{E}_{20}$) | Stdev ($m\hat{E}_{20}$) | Mean ($n\hat{E}_{20}$) | Stdev ($n\hat{E}_{20}$) |
|---|---|---|---|---|---|---|
| $f_1$ | $6.29 \times 10^{-1}$ | $3.71 \times 10^{-2}$ | $2.14 \times 10^{-1}$ | $7.44 \times 10^{-2}$ | $2.82 \times 10^{-1}$ | $3.24 \times 10^{-2}$ |
| $f_2$ | $6.58 \times 10^{-1}$ | $4.72 \times 10^{-2}$ | $7.55 \times 10^{-3}$ | $7.01 \times 10^{-3}$ | $5.76 \times 10^{-1}$ | $6.93 \times 10^{-2}$ |
| $f_3$ | $9.12 \times 10^{-1}$ | $2.35 \times 10^{-2}$ | $1.38 \times 10^{-1}$ | $1.98 \times 10^{-2}$ | $7.31 \times 10^{-1}$ | $3.20 \times 10^{-2}$ |
| $f_4$ | $5.97 \times 10^{-1}$ | $3.31 \times 10^{-2}$ | $1.63 \times 10^{-1}$ | $4.16 \times 10^{-2}$ | $4.29 \times 10^{-1}$ | $2.91 \times 10^{-2}$ |
| $f_5$ | $1.29 \times 10^{-1}$ | $2.92 \times 10^{-2}$ | $0.00 \times 10^{0}$ | $0.00 \times 10^{0}$ | $1.29 \times 10^{-1}$ | $2.92 \times 10^{-2}$ |
| $f_6$ | $3.24 \times 10^{-1}$ | $6.80 \times 10^{-2}$ | $1.39 \times 10^{-2}$ | $1.38 \times 10^{-2}$ | $3.47 \times 10^{-1}$ | $5.91 \times 10^{-2}$ |
| $f_7$ | $8.12 \times 10^{-1}$ | $3.21 \times 10^{-2}$ | $1.45 \times 10^{-1}$ | $7.05 \times 10^{-2}$ | $3.76 \times 10^{-1}$ | $5.05 \times 10^{-2}$ |
| $f_8$ | $5.34 \times 10^{-1}$ | $4.06 \times 10^{-2}$ | $1.29 \times 10^{-3}$ | $1.20 \times 10^{-3}$ | $5.27 \times 10^{-1}$ | $3.66 \times 10^{-2}$ |
| $f_9$ | $5.14 \times 10^{-1}$ | $5.78 \times 10^{-2}$ | $9.38 \times 10^{-2}$ | $3.82 \times 10^{-2}$ | $3.18 \times 10^{-1}$ | $3.11 \times 10^{-2}$ |

To calculate the standard entropy measure using $\hat{E}_{20}$, the values of each dimension of particle positions are placed in 20 bins. The computation involves sampling in those 20 bins in each dimension. For the work reported here, there were 100 particles and, apart from the Inverted Rosenbrock function which was investigated in four dimensions, all of the other problems were investigated in two dimensions. Since the nature of niching algorithms is to spread particles across the search space in an attempt to locate multiple optima, it was expected that sampling across the four and two dimensions in 20 bins would result in high diversity. This diversity would decrease gradually as particles started to converge towards optima but would still be maintained at high levels. As such, the results shown in Figures 11 and 12 mirror what was expected.

For the $m\hat{E}_{20}$, particle positions in the mentioned dimensions are also placed in 20 bins. However, since particles in a niche are likely to have converged, or are very close to each other, the diversity is expected to be low and tends to zero as particles converge towards the optima. Results shown are thus as expected.

In the case of $n\hat{E}_{20}$, the computation is similar to the standard $\hat{E}_{20}$ with the exception that only unique solutions are used. As such, a diversity measure using $n\hat{E}_{20}$ will be high, but not as high as that obtained using $\hat{E}_{20}$. The obtained results correctly predict this outcome.

Various research [12–17] has shown that entropy is a suitable measure to quantify dispersion. The work presented here postulates that entropy is a suitable measure for quantifying dispersion for solutions and candidate solutions of niching algorithms.

### 4.6. Solow–Polasky Diversity

Figures 13 and 14 and Table 7 show the diversity results as computed using the Solow–Polasky diversity measure, i.e., standard SPD, modified SPD (mSPD), and the niche diversity (nSPD).

As shown in Equation (9), the SPD is computed based on an $n \times n$ matrix, $M$, with entries $m_{i,j} = \exp(-\theta \|\mathbf{x}_i - \mathbf{x}_j\|_2)$. For the standard SPD, the entries are therefore the exponent of the pairwise distance of each particle to another particle, normalised using a user defined parameter, $\theta$. In this work, $\theta = 1$. Following this, the expectation is that the computed diversity will gradually decrease but stagnate at high values when particles converge to the location of the optima. The obtained results are therefore as expected.

For the mSPD, the entries $m_{i,j} = \exp(-\theta \|\mathbf{x}_i - \mathbf{x}_j\|_2)$ are computed from the particles in each niche and then an average is calculated. Since it is expected for the intra-niche distances to be small as particles converge towards an optimum, the obtained diversity should be low and tend towards zero. The results shown are therefore as expected.

The nSPD is computed similarly to the standard SPD, but with the exception that only the unique solutions are used. As such, whereas the distances for each of the unique solutions are the same as in SPD, the matrix is smaller because the number of niches is less. As such, the nSPD values will be smaller than those of the standard SPD, but the computed diversity will still be high. Results obtained from the experiment are therefore as expected.

The results obtained here concurs with [4] on the suitability of using the SPD measure to compute diversity of solutions and candidate solutions of niching algorithms.

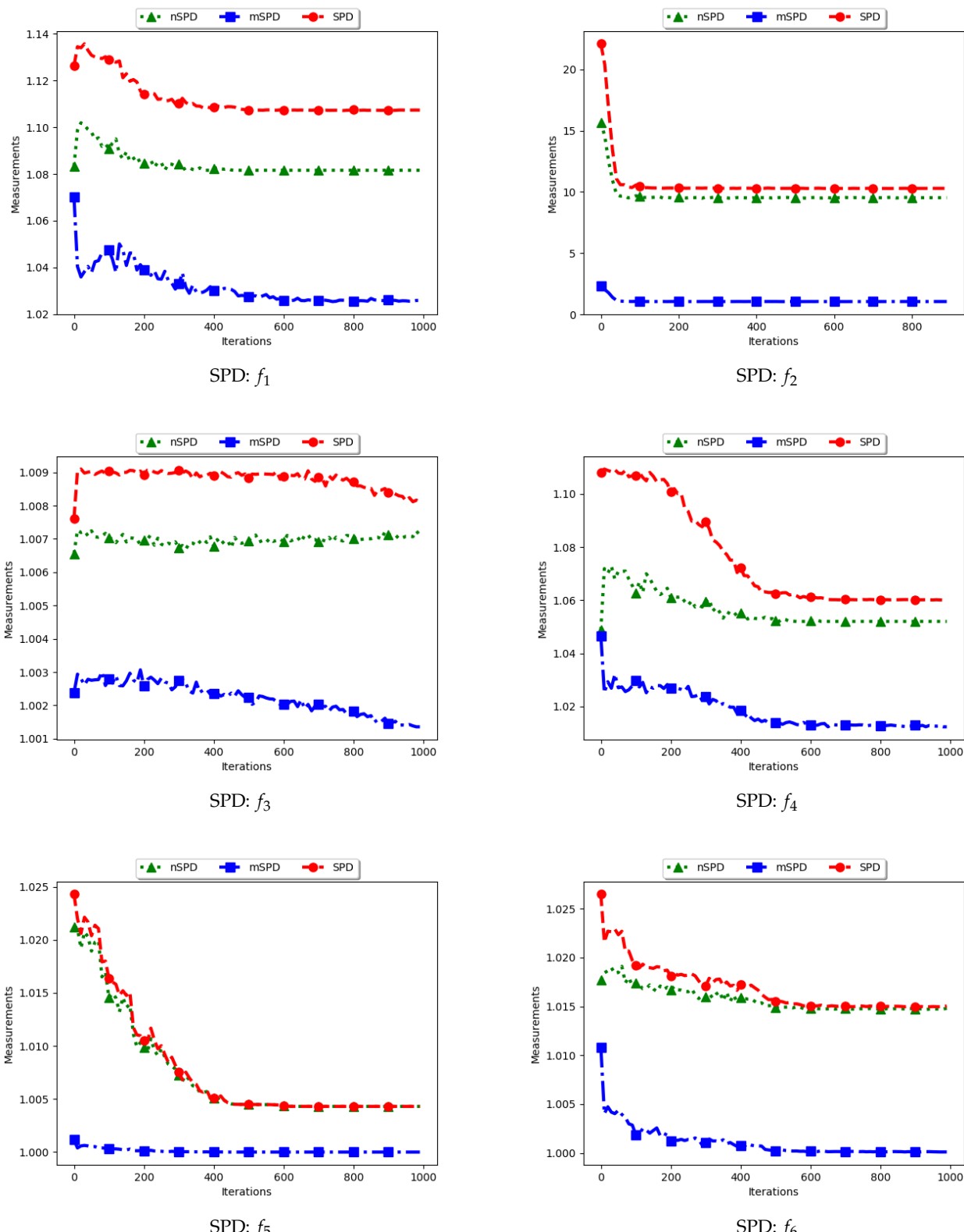

**Figure 13.** Quantification of diversity using the Solow–Polasky Diversity.

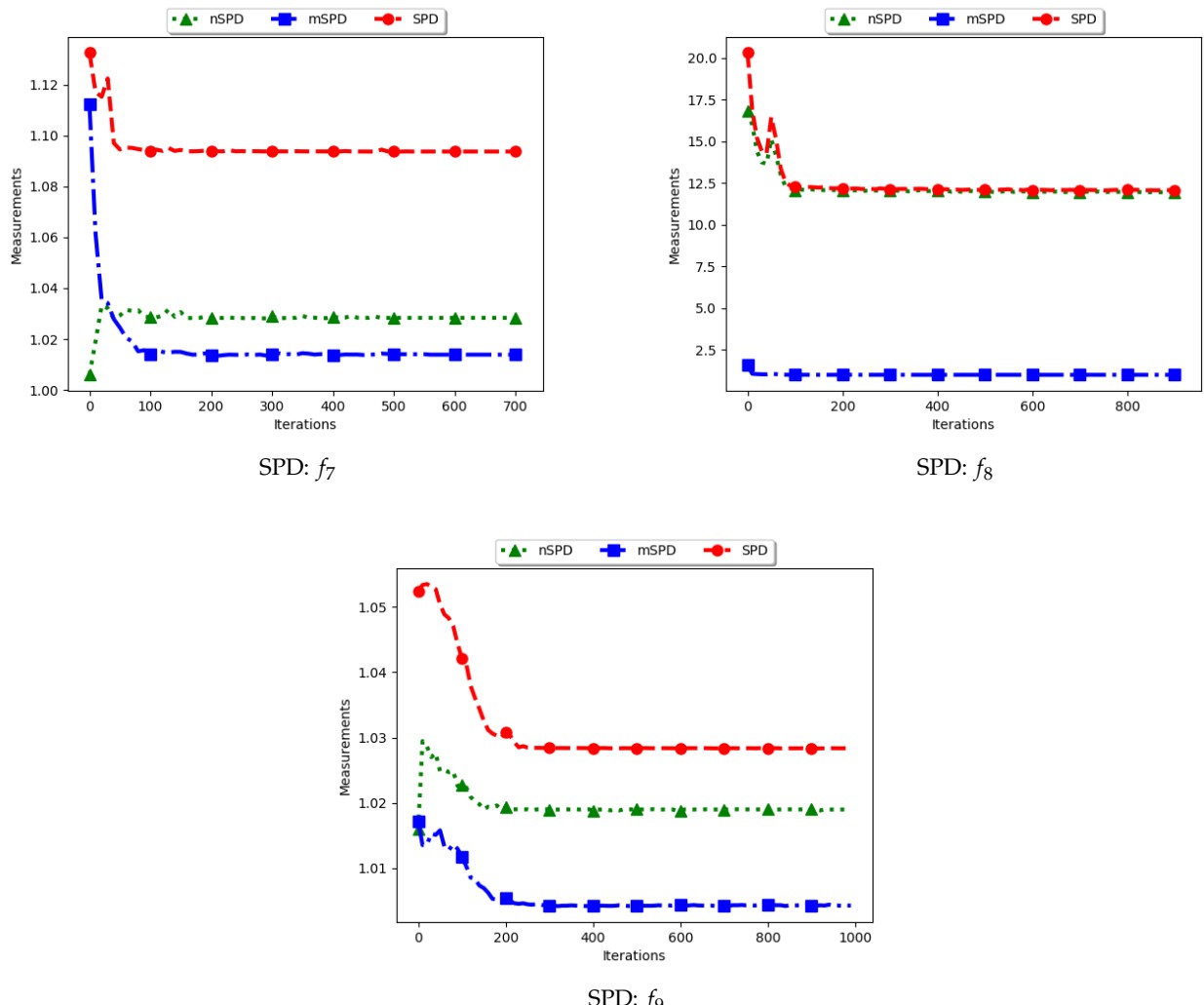

SPD: $f_7$

SPD: $f_8$

SPD: $f_9$

**Figure 14.** Quantification of diversity using the Solow–Polasky Diversity (*Cont.*).

**Table 7.** Mean ($\mu$) and standard deviation ($\sigma$) for the SPD measure and its variants at the last iteration.

| Function | Mean (SPD) | Stdev (SPD) | Mean (mSPD) | Stdev (mSPD) | Mean (nSPD) | Stdev (nSPD) |
|---|---|---|---|---|---|---|
| $f_1$ | $1.11 \times 10^0$ | $6.63 \times 10^{-3}$ | $1.03 \times 10^0$ | $1.05 \times 10^{-2}$ | $1.08 \times 10^0$ | $1.30 \times 10^{-4}$ |
| $f_2$ | $1.03 \times 10^1$ | $1.35 \times 10^0$ | $1.04 \times 10^0$ | $3.99 \times 10^{-2}$ | $9.52 \times 10^0$ | $1.27 \times 10^0$ |
| $f_3$ | $1.01 \times 10^0$ | $3.34 \times 10^{-4}$ | $1.00 \times 10^0$ | $1.88 \times 10^{-4}$ | $1.01 \times 10^0$ | $1.91 \times 10^{-4}$ |
| $f_4$ | $1.06 \times 10^0$ | $4.19 \times 10^{-3}$ | $1.01 \times 10^0$ | $3.49 \times 10^{-3}$ | $1.05 \times 10^0$ | $5.76 \times 10^{-5}$ |
| $f_5$ | $1.00 \times 10^0$ | $3.49 \times 10^{-3}$ | $1.00 \times 10^0$ | $0.00 \times 10^0$ | $1.00 \times 10^0$ | $3.49 \times 10^{-3}$ |
| $f_6$ | $1.01 \times 10^0$ | $1.87 \times 10^{-3}$ | $1.00 \times 10^0$ | $2.41 \times 10^{-4}$ | $1.01 \times 10^0$ | $1.47 \times 10^{-3}$ |
| $f_7$ | $1.09 \times 10^0$ | $6.42 \times 10^{-3}$ | $1.01 \times 10^0$ | $6.84 \times 10^{-3}$ | $1.03 \times 10^0$ | $6.30 \times 10^{-3}$ |
| $f_8$ | $1.21 \times 10^1$ | $1.31 \times 10^0$ | $1.01 \times 10^0$ | $5.51 \times 10^{-3}$ | $1.19 \times 10^1$ | $1.28 \times 10^0$ |
| $f_9$ | $1.03 \times 10^0$ | $4.96 \times 10^{-3}$ | $1.00 \times 10^0$ | $1.96 \times 10^{-3}$ | $1.02 \times 10^0$ | $6.19 \times 10^{-4}$ |

## 4.7. Swarm Radius

Figures 15 and 16 as well as Table 8 show the results of diversity as quantified using the standard swarm radius, SR, modified SR (mSR) and niche diversity (nSR).

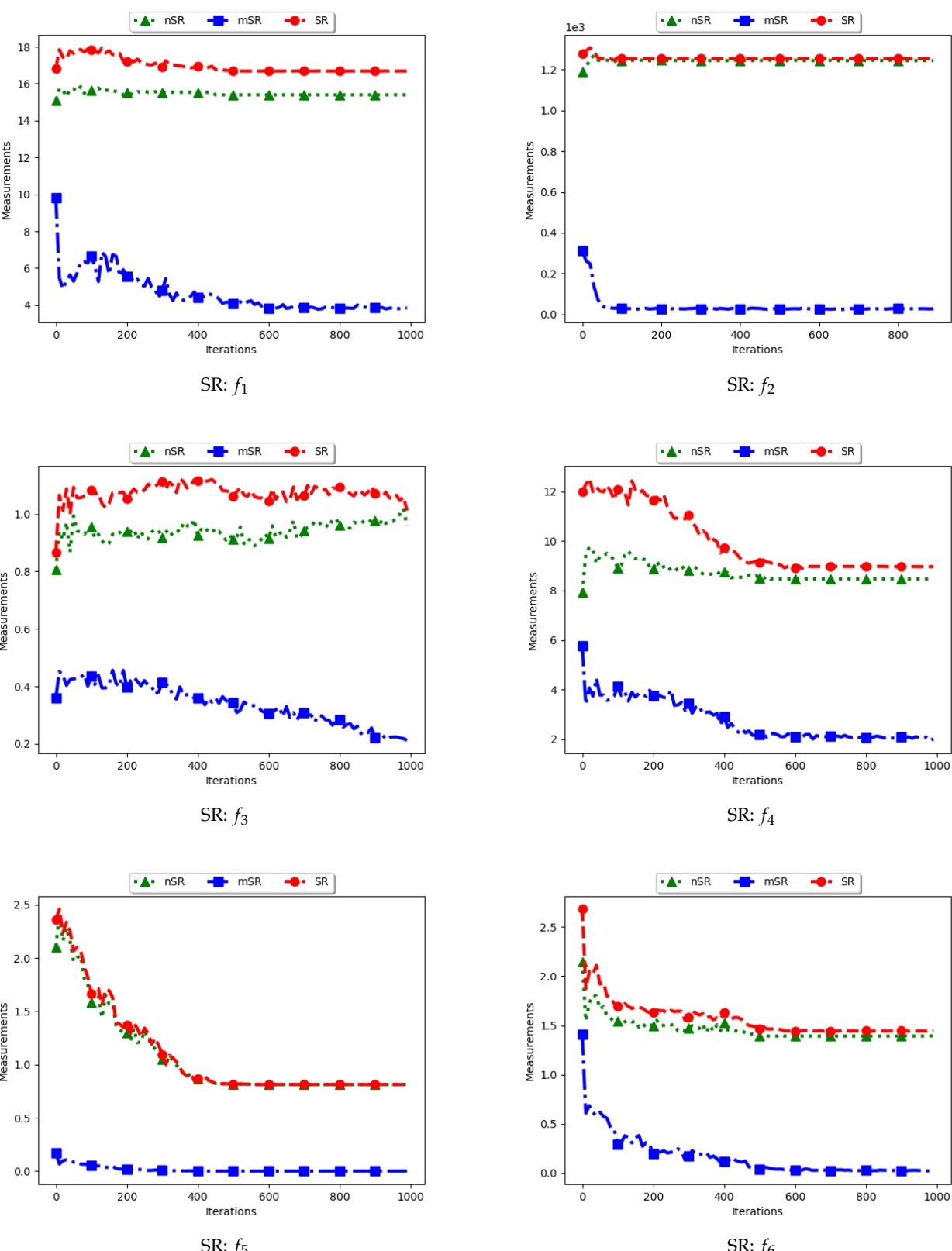

**Figure 15.** Quantification of diversity using the Swarm Radius.

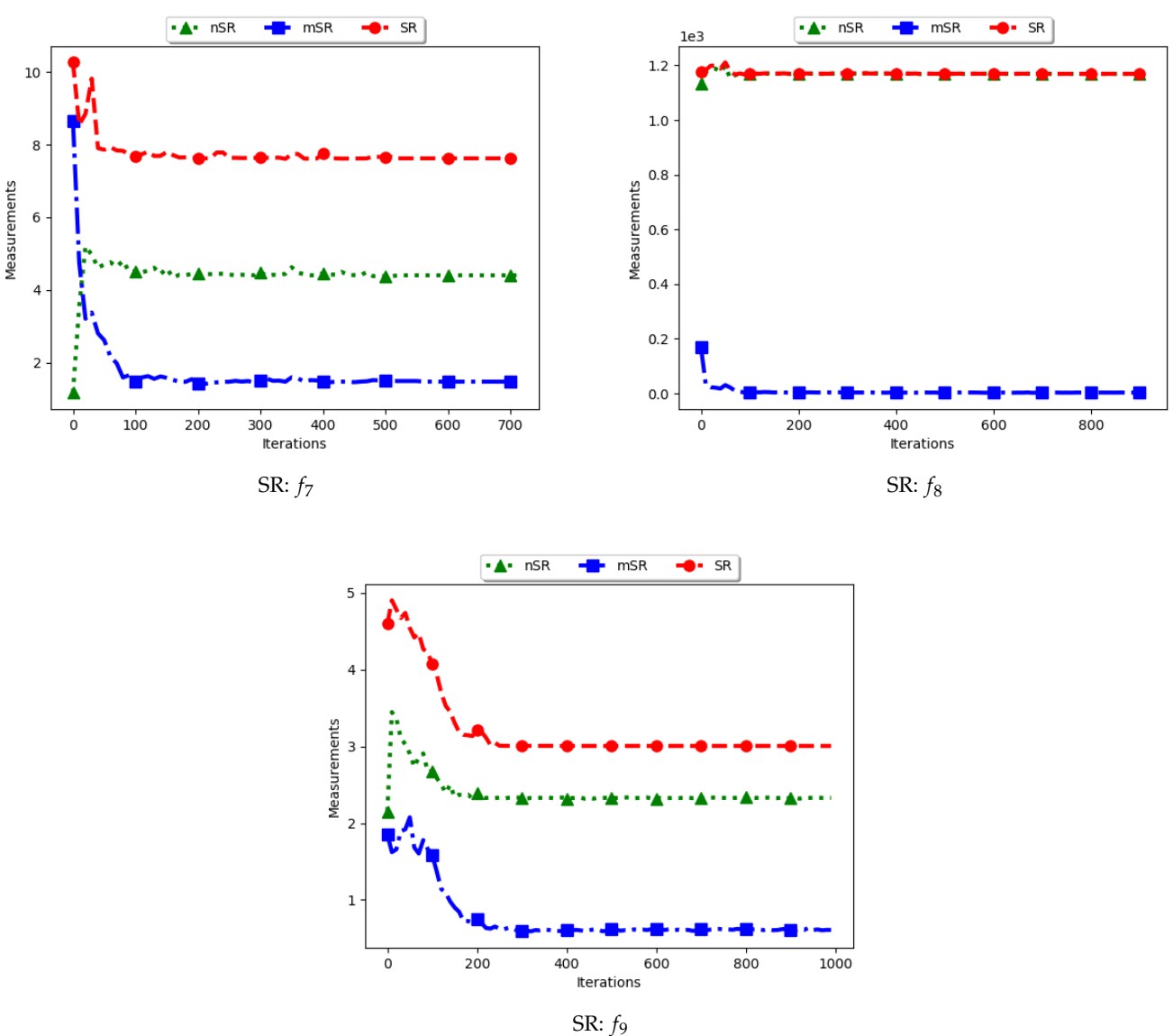

**Figure 16.** Quantification of diversity using the Swarm Radius (*Cont.*).

**Table 8.** Mean ($\mu$) and standard deviation ($\sigma$) for SR measure and its variants at the last iteration.

| Function | Mean (SR) | Stdev (SR) | Mean (mSR) | Stdev (mSR) | Mean (nSR) | Stdev (nSR) |
|----------|-----------|------------|------------|-------------|------------|-------------|
| $f_1$ | $1.67 \times 10^1$ | $1.44 \times 10^0$ | $3.81 \times 10^0$ | $1.63 \times 10^0$ | $1.54 \times 10^1$ | $1.40 \times 10^0$ |
| $f_2$ | $1.25 \times 10^3$ | $4.04 \times 10^1$ | $2.62 \times 10^1$ | $2.99 \times 10^1$ | $1.24 \times 10^3$ | $3.19 \times 10^1$ |
| $f_3$ | $9.95 \times 10^{-1}$ | $1.41 \times 10^{-1}$ | $1.50 \times 10^{-1}$ | $3.32 \times 10^{-2}$ | $9.84 \times 10^{-1}$ | $1.31 \times 10^{-1}$ |
| $f_4$ | $8.96 \times 10^0$ | $7.36 \times 10^{-1}$ | $2.10 \times 10^0$ | $5.99 \times 10^{-1}$ | $8.46 \times 10^0$ | $2.15 \times 10^{-1}$ |
| $f_5$ | $8.13 \times 10^{-1}$ | $6.34 \times 10^{-1}$ | $0.00 \times 10^0$ | $0.00 \times 10^0$ | $8.13 \times 10^{-1}$ | $6.34 \times 10^{-1}$ |
| $f_6$ | $1.44 \times 10^0$ | $2.01 \times 10^{-1}$ | $2.27 \times 10^{-2}$ | $4.81 \times 10^{-2}$ | $1.39 \times 10^0$ | $1.08 \times 10^{-1}$ |
| $f_7$ | $7.62 \times 10^0$ | $6.12 \times 10^{-1}$ | $1.47 \times 10^0$ | $7.20 \times 10^{-1}$ | $4.40 \times 10^0$ | $1.07 \times 10^0$ |
| $f_8$ | $1.17 \times 10^3$ | $6.37 \times 10^1$ | $3.30 \times 10^0$ | $3.64 \times 10^0$ | $1.17 \times 10^3$ | $6.37 \times 10^1$ |
| $f_9$ | $3.00 \times 10^0$ | $5.31 \times 10^{-1}$ | $6.07 \times 10^{-1}$ | $3.04 \times 10^{-1}$ | $2.33 \times 10^0$ | $7.54 \times 10^{-2}$ |

For the SR measure, both the standard SR and the nSR were computed using the "swarm best", i.e., the solution with the best objective function value. Since the goal of niching algorithms is to obtain multiple optima, the SR measure obtained the largest distance between any of the "swarm best" and a candidate solution. Since each candidate solution is optimising an optimum, the expectation is that the diversity quantified using SR will monotonically decrease as candidate solutions converge. This is because the diversity is not measured on how far apart the solutions are from each other but from the "swarm best" which are multiple. Following this, where SR reports high diversity, it is likely to be as a result of the presence of outliers. This is the case with the obtained results.

In the case of mSR, the distances are computed with respect to each niche, and therefore the neighbourhood best is used. Diversity is thus expected to be low and tending towards zero as particles converge towards an optimum. As such, the achieved results are as expected.

For the nSR, unique solutions are used. As such, the obtained results will be somewhat similar to those of the standard SR. This is because, for both cases, the particle with the best objective function value is used. Where two candidate solutions have the same objective function value, the computed distance is zero. Where the nSR value is smaller than SR, the likely explanation is that the furthest particle among the set of unique solutions was closer to a solution than in the general swarm. This too is expected.

These results mirror those of the ADSC. For both cases, the distances are calculated from a candidate solution with the best objective fitness. For niching algorithms, the presence of multiple optima for which the distances can be calculated from means that lower diversity will be expected. As such, the only explanation to the reported high diversity is likely to be the presence of outliers. A similar result was reported in [11].

### 4.8. Swarm Diameter

Figures 17 and 18 as well as Table 9 show the diversity results as computed using the swarm diameter measures, i.e., standard SDM, modified SDM (mSDM) and the niche diversity (nSDM).

For the SDM, the computation is geared towards the maximum distance between any two particles/candidate solutions. As such, the expectation is that standard SDM will obtain high diversity during both the exploration and the exploitation phases. The obtained results are therefore as expected.

The mSDM is only computed per each niche and then an average is calculated. As such, it is expected that the diversity will be small and tending towards zero as convergence towards the location of optima is achieved. As such, results of the mSDM are as expected.

The nSDM is computed in a similar manner to that of the SDM. The only difference is that the nSDM is computed using only unique solutions. As such, the nSDM is expected to show a high diversity both at the exploration and exploitation phases.

In summary, whereas the SDM measure can truly reflect how far apart both the solutions and candidate solutions are, these solutions are likely to also contain outliers. The high diversity as shown may thus not be a measure of how truly diverse the obtained solutions are.

**Table 9.** Mean ($\mu$) and standard deviation ($\sigma$) for SDM measure and its variants at the last iteration.

| Function | Mean (SDM) | Stdev (SDM) | Mean (mSDM) | Stdev (mSDM) | Mean (nSDM) | Stdev (nSDM) |
|---|---|---|---|---|---|---|
| $f_1$ | $1.86 \times 10^1$ | $5.09 \times 10^{-1}$ | $4.82 \times 10^0$ | $1.97 \times 10^0$ | $1.59 \times 10^1$ | $3.25 \times 10^{-2}$ |
| $f_2$ | $1.28 \times 10^3$ | $4.07 \times 10^1$ | $2.63 \times 10^1$ | $3.00 \times 10^1$ | $1.25 \times 10^3$ | $2.79 \times 10^1$ |
| $f_3$ | $1.17 \times 10^0$ | $6.38 \times 10^{-2}$ | $1.65 \times 10^{-1}$ | $3.62 \times 10^{-2}$ | $1.13 \times 10^0$ | $2.36 \times 10^{-2}$ |
| $f_4$ | $9.54 \times 10^0$ | $9.28 \times 10^{-1}$ | $2.38 \times 10^0$ | $6.42 \times 10^{-1}$ | $8.59 \times 10^0$ | $1.04 \times 10^{-2}$ |
| $f_5$ | $8.45 \times 10^{-1}$ | $6.72 \times 10^{-1}$ | $0.00 \times 10^0$ | $0.00 \times 10^0$ | $8.45 \times 10^{-1}$ | $6.72 \times 10^{-1}$ |
| $f_6$ | $2.43 \times 10^0$ | $3.61 \times 10^{-1}$ | $2.27 \times 10^{-2}$ | $4.81 \times 10^{-2}$ | $2.38 \times 10^0$ | $2.76 \times 10^{-1}$ |
| $f_7$ | $1.28 \times 10^1$ | $9.33 \times 10^{-1}$ | $2.06 \times 10^0$ | $1.01 \times 10^0$ | $4.95 \times 10^0$ | $1.04 \times 10^0$ |
| $f_8$ | $1.22 \times 10^3$ | $6.35 \times 10^1$ | $3.30 \times 10^0$ | $3.64 \times 10^0$ | $1.22 \times 10^3$ | $6.37 \times 10^1$ |
| $f_9$ | $4.94 \times 10^0$ | $8.02 \times 10^{-1}$ | $7.74 \times 10^{-1}$ | $3.43 \times 10^{-1}$ | $3.73 \times 10^0$ | $9.46 \times 10^{-2}$ |

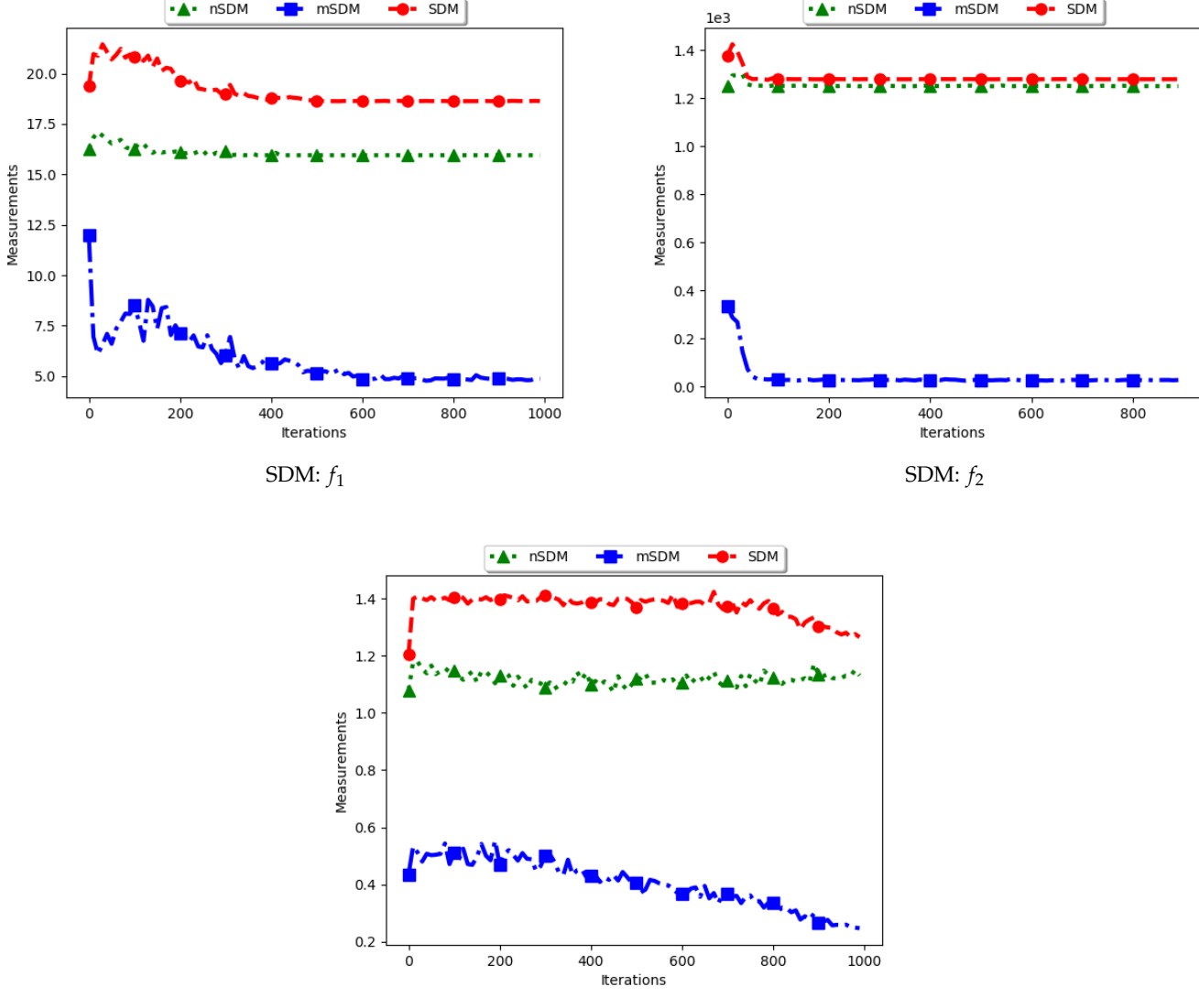

SDM: $f_1$

SDM: $f_2$

SDM: $f_3$

**Figure 17.** Quantification of diversity using the Swarm Diameter.

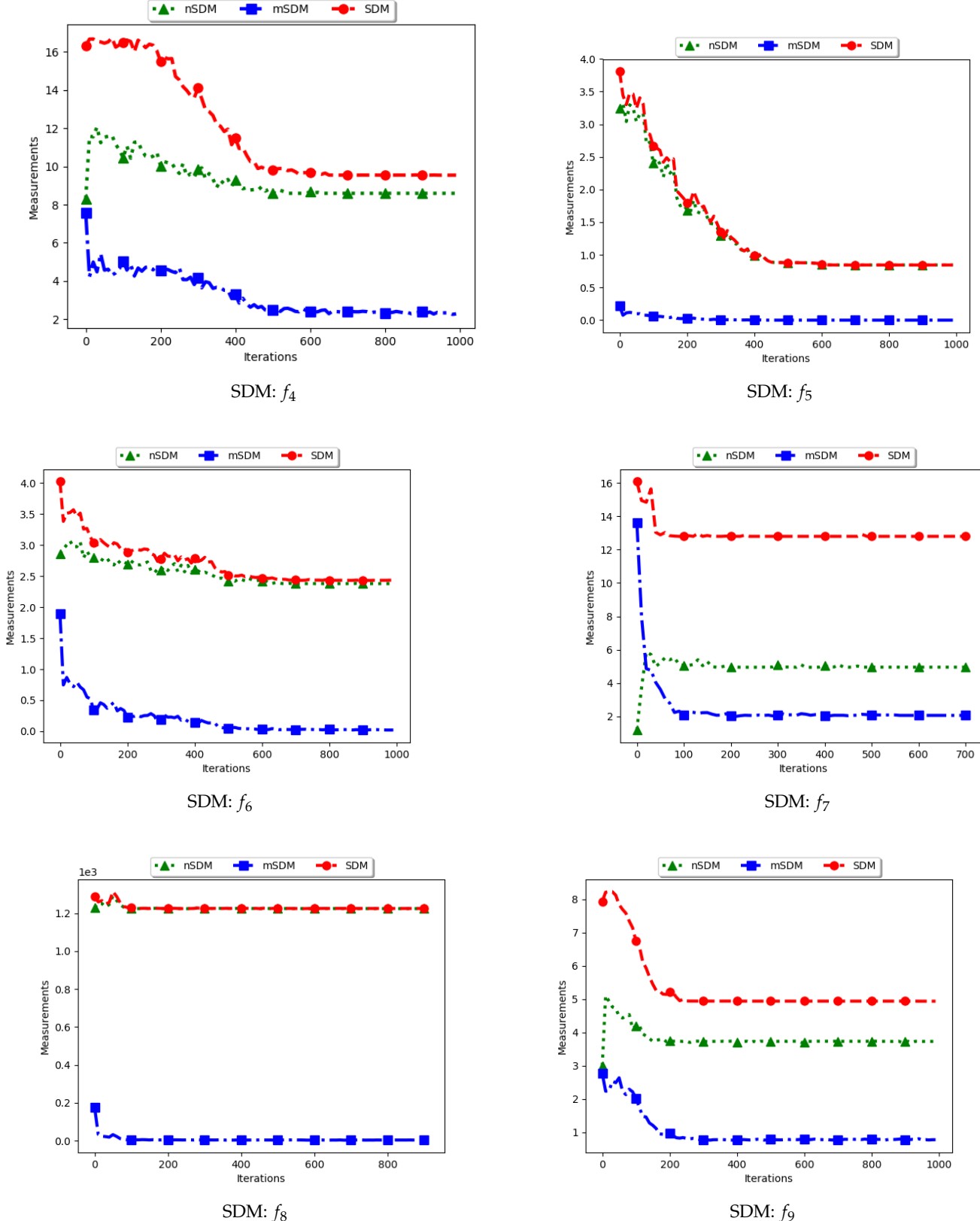

**Figure 18.** Quantification of diversity using the Swarm Diameter(*Cont.*).

## 5. Conclusions

This paper offers a concise understanding of diversity measures for quantifying the spread of candidate solutions and solutions of a niching algorithm. The paper does not attempt to compare the diversity measures, but instead analyse whether they can correctly quantify diversity when used with niching algorithms. The diversity measures are discussed with respect to the search space, i.e., swarm diversity, and with respect to solution space, i.e., niche diversity.

An empirical study involving the reviewed measures was carried out using a set of multimodal functions optimised using the enhanced species-based particle swarm optimisation (ESPSO) niching algorithm. The obtained diversity results showed that some measures are more suited to the task than others. However, the conclusions were drawn as remarks for each diversity measure rather than a conclusive determination on which diversity measure is better than the other.

The discussion shows that both swarm diversity and niche diversity are important for the purpose of niching algorithms. The calculation of swarm diversity, by first subjecting the measures to each of the candidate niches, is critical because niching algorithms cluster candidate solutions during the search process. The niche diversity is important because it shows diversity of the unique solutions ("would-be solutions") during exploration and during exploitation. High niche diversity during the exploration phase indicates that a niching algorithm is capable of identifying potential niches, while high diversity at the exploitation phase means that the algorithm is able to maintain the found niches. It is, however, expected that niche diversity will decrease as the swarm moves from the exploration phase to the exploitation phase.

Diversity is an important aspect of a population-based algorithm because it determines the performance over a given set of problems. It is thus important to investigate diversity measures that can be utilised to analyse the spread of candidate solutions. The presented study can thus only be seen as a first step towards this investigation.

**Author Contributions:** The conceptualization of this work was carried out by J.M. and A.P.E. The two authors also devised the methodology for the work; J.M. and F.V.N. designed the experiments and F.V.N. was pivotal in both software development and the validation of the experiments. The experiments' formal analysis and investigation were carried out by J.M. In addition, J.M. wrote the original draft of the paper. The review and editing of the work was carried out by J.M. and A.P.E. The visualization of the experiment results was carried out by J.M. and F.V.N., whereas the overall supervision of the work was carried out by A.P.E. Furthermore, J.M. and A.P.E. were involved in funding acquisition. All authors have read and agreed to the published version of the manuscript.

**Funding:** This work is based on the research supported by the National Research Foundation (NRF) of South Africa (Grant Number 89630). The opinions, findings and conclusions or recommendations expressed in this article is that of the author(s) alone, and not that of the NRF.

**Acknowledgments:** The authors would like to thank the Centre for High Performance Computing (CHPC) based in Cape Town, South Africa for allowing the use of their servers to run the simulation for the experiments reported in this paper. The authors would also like to thank the entire development team of Computation Intelligence Library (CILib) based in South Africa.

**Conflicts of Interest:** The authors declare no conflict of interest.

## Abbreviations

The following abbreviations are used in this manuscript:

PSO    Particle Swarm Optimisation
DE    Differential Evolution
GA    Genetic Algorithms
ESPO    Enhanced Species-Based Particle Swarm Optimisation
SD    Sum of Distances
SDNN    Sum of Distances to Nearest Neighbour
ADSC    Average Distance Around the Swarm Centre
ADAA    Average of the Mean Distance around all Candidate Solutions
SPD    Solow–Polasky Diversity
SDM    Swarm Diameter
SR    Swarm Radius

## Appendix A. Branin

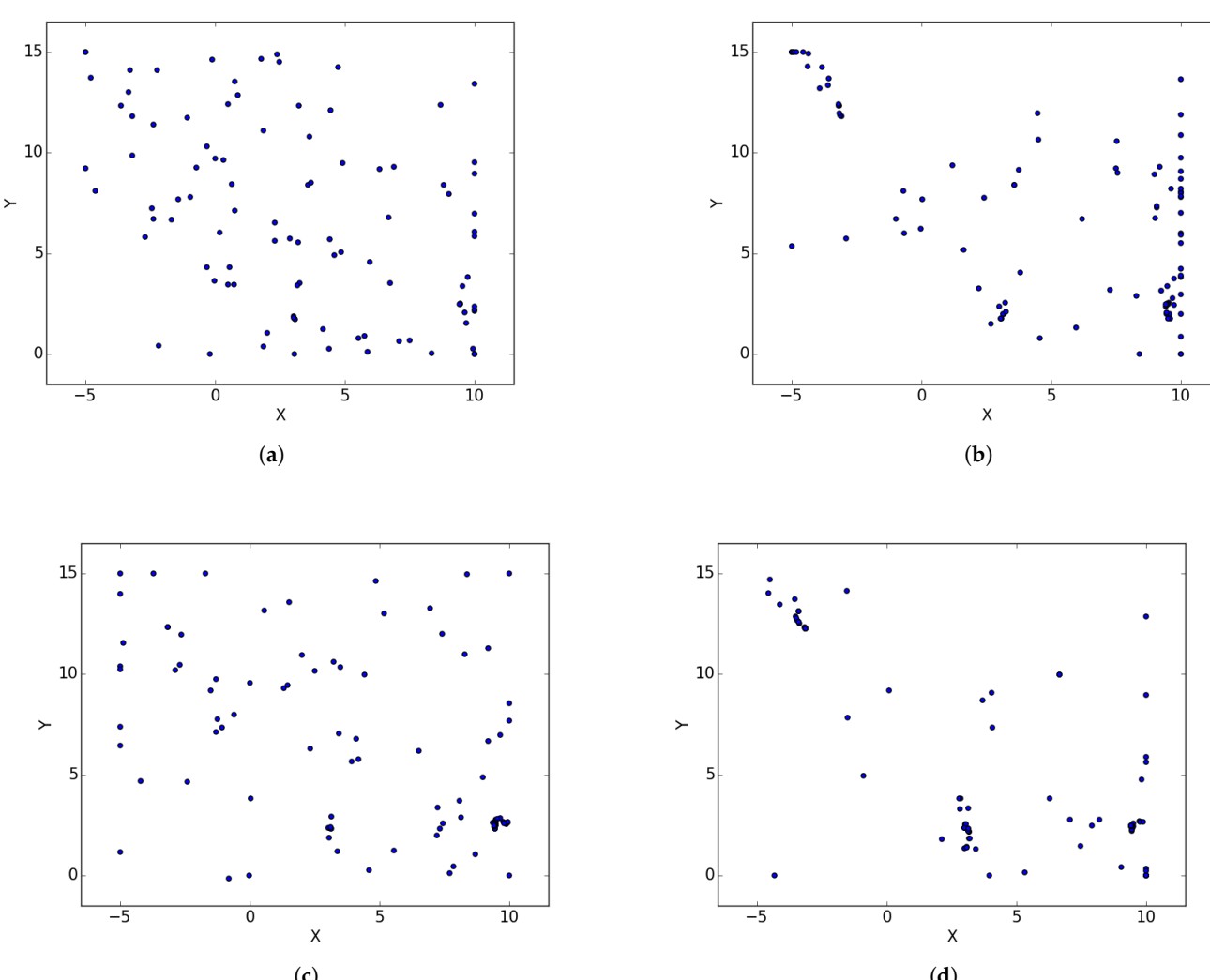

**Figure A1.** *Cont.*

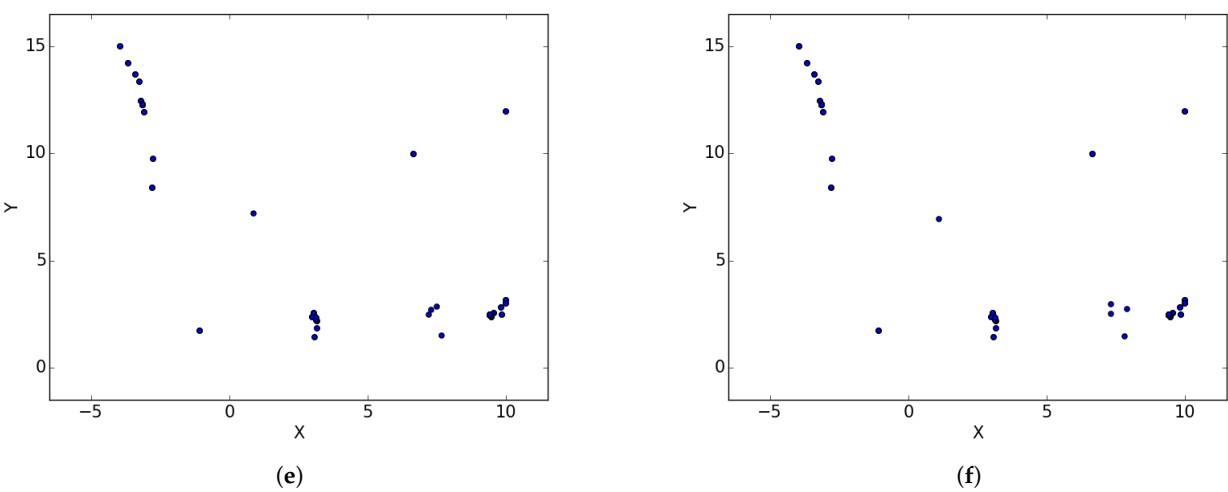

(e)

(f)

**Figure A1.** Branin Rcos at different iterations. (**a**) $f_1$: iteration 50; (**b**) $f_1$: iteration 100; (**c**) $f_1$: iteration 250; (**d**) $f_1$: iteration 500; (**e**) $f_1$: iteration 750; (**f**) $f_1$: iteration 1000.

## Appendix B. EggHolder

(a)

(b)

(c)

(d)

**Figure A2.** *Cont.*

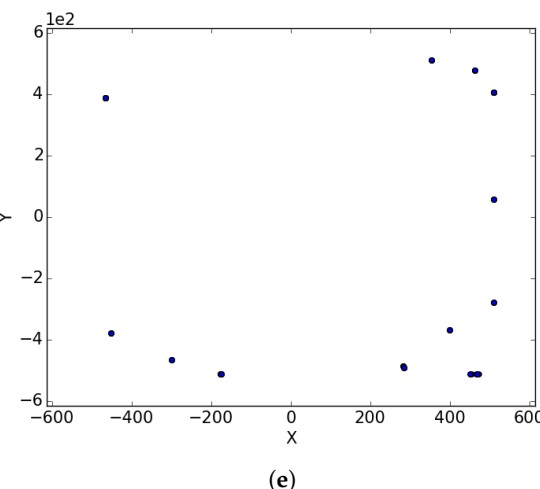

(e)

**Figure A2.** Inverted Egg Holder at different iterations. (**a**) $f_2$: iteration 50; (**b**) $f_2$: iteration 100; (**c**) $f_2$: iteration 250; (**d**) $f_2$: iteration 500; (**e**); $f_2$: iteration 750.

## Appendix C. Inverted Schwefel Problem 2_26

<table>
<tr><td align="center">(a)</td><td align="center">(b)</td></tr>
<tr><td align="center">(c)</td><td align="center">(d)</td></tr>
</table>

**Figure A3.** *Cont.*

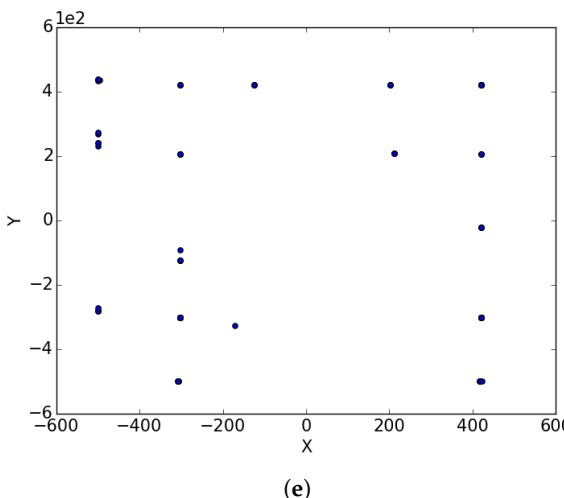

(**e**)

**Figure A3.** Inverted Schwefel Problem 2_26 at different iterations. (**a**) $f_8$: iteration 50; (**b**) $f_8$: iteration 100; (**c**) $f_8$: iteration 250; (**d**) $f_8$: iteration 500; (**e**) $f_8$: iteration 750.

## Appendix D. Test Functions in 2D

(**a**)

(**b**)

(**c**)

(**d**)

**Figure A4.** *Cont.*

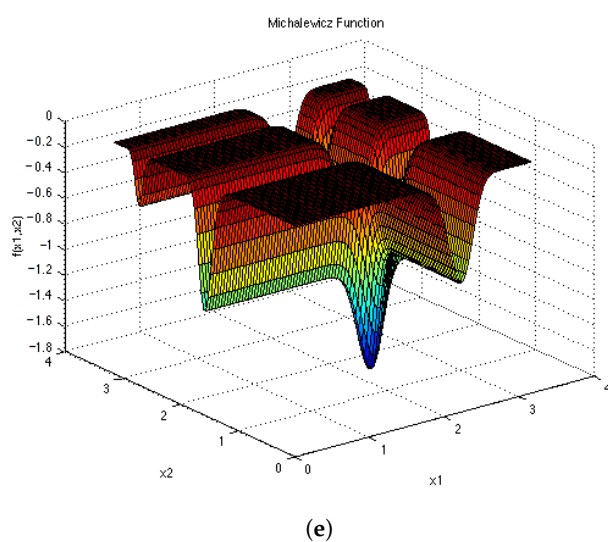

(**e**)

**Figure A4.** Test Functions shown in 2D. (**a**) $f_1$; (**b**) $f_2$; (**c**) $f_3$; (**d**) $f_4$; (**e**) $f_5$.

## Appendix E. Test Functions in 2D

(**a**)

(**b**)

(**c**)

**Figure A5.** *Cont*.

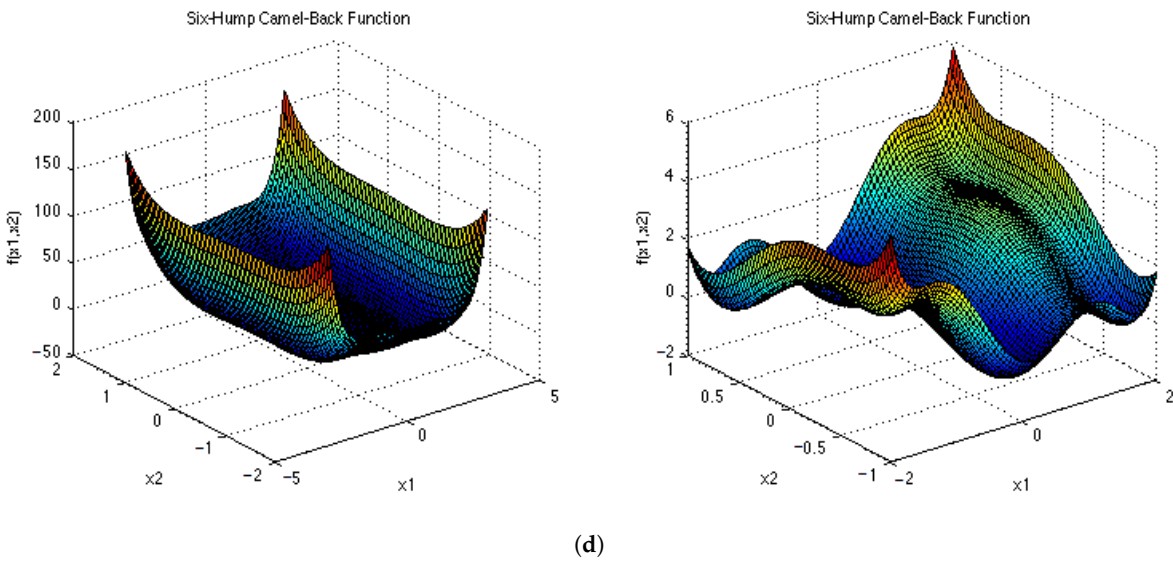

(**d**)

**Figure A5.** Test Functions shown in 2D. (**a**) $f_6$; (**b**) $f_7$; (**c**) $f_8$; (**d**) $f_9$.

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
