# Peer review of "Diversity Measures for Niching Algorithms"

_algorithms, doi:10.3390/a14020036_

Round 1
Reviewer 1 Report
The article is excellent and relevant; the authors present an interesting proposal for Diversity Measures for Niching Algorithms. The authors address an important topic in the development of algorithms based on swarms of particles. In general, the article displays a suitable development considering a qualitative analysis.
Mayor revision
It is understood that a direct comparison between algorithms is not being made; however, it would be suitable to include statistical results (tables) which allow the observations made in each experimental case; this might be done because the initialization of the particles is random; therefore statistical results of the diversity measurements could be obtained. In the case that the authors think that it is not possible, please adequately justify this.
Minor revision
- For equations (1)-(6) and (10)-(12) it is suggested to use the \substack instruction (LaTex) to organize the lower limits of the summation in two lines.
- Try to improve the figure 1 to better display the concept of Swarm Diversity; moreover, intermediate states could be included to show how the position of the particles could evolve.
- Consider including a figure with the graphical representation of the test functions used (in two dimensions).
- According to Turniting, few texts are similar to the presented in [Mwaura2016] (only 8%) then is suggested to modify these few texts. The paragraphs to adjust are:
‘‘Numerous niching algorithms have been developed to find multiple optima for multimodal optimisation problems’’
‘‘This paper provides a critical review of diversity measures’’
‘‘Multimodal problems may have anything from a few to a very large number of optima. The goal of niching algorithms is to locate as many global and/or local optima as possible that are considered satisfactory [3] and that are as diverse as possible [4].’’
‘‘The sum of distances (SD) measure computes how diverse the candidate solutions are, by finding the square root of the sum of their distances from one another [4]. The SD is calculated as’’
‘‘If the SD measure is used to quantify swarm diversity of a niching algorithm, an incorrect impression about diversity will be obtained. This is because SD values will be high if inter-niche2 distances are large, despite the fact that particles within each niche may be very close to the corresponding optimum, that is, small intra-niche3 distances. Note that, for niching algorithms, as intra-niche distances approach zero, the niching algorithm starts to converge with the expectation that the swarm diversity measure should approach zero.’’
‘‘The sum of distances to the nearest neighbour (SDNN) [8] quantifies diversity of the candidate solutions by summing the distances of each candidate solution to its nearest neighbour [9].’’
‘‘When SDNN is employed to measure the swarm diversity of a niching algorithm, the DSDNN may not follow the expected trend of converging to zero as niches start to converge. The main reason for this is that the nearest neighbour (NN) of a particle might be a particle in a niche that has converged. This means that the NNs will cause high diversity, while the actual diversity is good due to niches having converged.’’
‘‘The Solow-Polasky diversity (SPD) measure [18] was recently proposed by Preuss and Wessing [4] to calculate diversity when using niching algorithms. To calculate SPD for the candidate solutions, an n x n matrix, M, has to be constructed. The entries of the matrix are defined as’’
‘‘The main drawback of this measure is in its reliance on the inverse of the matrix. In situations where an inverse cannot be obtained, such as when the matrix is singular, the measure cannot be calculated.’’
‘‘The parameters of the ESPSO algorithm were tuned using the iterated F-Race [19] algorithm. To carry out the tuning, the algorithm parameters were tuned for each of the multimodal problems listed in Table 1. Since there are different parameters used for each of the multimodal problems, the parameters are not listed in this paper.’’
‘‘For all functions, two dimensions were used. Each algorithm was applied to each function for 1000 iterations, for 30 independent runs. The population size was set to 100 and the particles that leave the search space were randomly re-initialised within the search domain.’’
‘‘This technique determines whether a candidate solution is unique by checking that, between a candidate solution, xi, and each other candidate solution, xj, the following holds (assuming maximisation):’’
References:
[Mwaura2016] Jonathan Mwaura, Andries P. Engelbrecht, Filipe V. Nepocumeno, Performance measures for niching algorithms, 2016 IEEE Congress on Evolutionary Computation (CEC), 2016.
Reviewer 2 Report
The paper looks at a number of diversity measures for point distributions via numerical simulations in 2 dimensions. Essentially they describe a number if diversity measures, both for the whole swarm, averaged over each niche, and between niches. These are calculated for 9 test problems, all in two dimensions, and graphed. I found their analysis of the graphed results was perfunctory and uninforming.
The authors do not say whether the same runs are tested using different diversities or if the runs are different for each diversity. The authors seem to implicitly assume that if a niche is formed then it is around a maximizer. For example on lines 4-6 of the abstract the authors write: ". . . while a low diversity means the candidates are clustered at optima." This affects all of their conclusions. The authors make no comment about why a diversity measure might be of use. The behaviour of the nXXX and mXXX diversities are strongly dependent on the process of forming the niches, which is not discussed. At no point do they report any information on the number of niches formed by ESPSO. The authors write as though diversity going to zero is good, but that can happen when ESPSO converges to a point far from any peak --- in that case low diversity is evidence the method has stalled.
line 157 and 163: by pw do you mean the number of particles in bin w divided by the total number of particles?
line 265: were the points from the same 30 runs used to calculate all diversity measures, or were different runs used for each measure?
Table 1: right hand column xi in [?,?]n the n should be deleted in each case. In this table you use n for problem dimension, in equation (8) you use D and n is used elsewhere for swarm size. In 2 dimensions the inverted Rosenbrock does not have local peaks: it has one global peak only. This needs to be reflected in your comments throughout the paper.
line 194: how nSD is calculated (and other nXXX measures) needs to be described much more clearly.
Some of the diversity measured are mathematically very simple. have you considered trying to derive some relationships between the XX, mXX, and nXX forms ?
Minor Comments:
line 48: VI should be IV
line 108: what does good mean here?
line 184: should high be maximum possible?
line 282: what shaded region ?? I can't see a shaded region.
line 290: converges should be converge
Figures 7 and 8: the captions are wrong.
Reviewer 3 Report
The paper compares several diversity measures through the run of a ESPSO algorithm, on several benchmark functions. The conclusions highlight a number of key differences between the measures, and the results do provide an informative insight into the different approaches. The paper is well written and covers an important topic.
I am a little wary of the fact that only one algorithm is covered by the experiments; there is a risk to generality because of this that could be mentioned. However, I do not think this is a big obstacle as it does allow better focus on comparison of the measures. A couple of other minor changes should also be made.
P9: "Figures 3 and 4...The shaded region at the bottom of each curve shows the number of solutions (unique solutions) found at each iteration as indicated by the right hand axis"; I do not see a shaded region or a right-axis in either figures 3 or 4.
All figures from 3: these are the results from 30 repeat runs. that's good, but it's not clear whether these figures show the mean or some other aggregation over all 30 runs. I think it might also be worthwhile adding the range over the 30 runs to these figures (error bars or feint lines would be enough); similar for later figures.
Round 2
Reviewer 1 Report
The authors have successfully addressed all the comments and the paper may be accepted for publication.
In addition I recommend use latex format $3.02\times10^{-6}$ instead of 3.02e-06 (example) in Tables 2-9.
Author Response
Comment: In addition I recommend use latex format $3.02\times10^{-6}$ instead of 3.02e-06 (example) in Tables 2-9.
Reply: This has now been accomplished. Please see the article.
Thank you
Reviewer 2 Report
Dear Authors,
Many thanks for your revised paper, and thank you for highlighting changes in blue. Could you please address the remaining points I raised in the first part of my original review. These points, which come before itemised points are the most important in that review.
In particular, how are niches defined. On line 117 you write: "Niching algorithms define niche formation differently. " As I have asked before, could you please describe, at least in broad brush terms, how you formed niches.
many thanks
a reviewer
Author Response
Comment: In particular, how are niches defined. On line 117 you write: "Niching algorithms define niche formation differently. " As I have asked before, could you please describe, at least in broad brush terms, how you formed niches.
Reply: Thank you for pointing this out. A paragraph has been added on sub-section 2.1.2 that narrates how ESPSO forms niches. This can be found between lines 117 -- 128
Round 3
Reviewer 2 Report
Many thanks for the niche forming description.